# Reliable learning in challenging environments

**Maria-Florina Balcan**
Carnegie Mellon University
`ninamf@cs.cmu.edu`

**Steve Hanneke**
Purdue University
`steve.hanneke@gmail.com`

**Rattana Pukdee**
Carnegie Mellon University
`rpukdee@cs.cmu.edu`

**Dravyansh Sharma**
Carnegie Mellon University
`dravyans@cs.cmu.edu`

## Abstract

The problem of designing learners that provide guarantees that their predictions are provably correct is of increasing importance in machine learning. However, learning theoretic guarantees have only been considered in very specific settings. In this work, we consider the design and analysis of reliable learners in challenging test-time environments as encountered in modern machine learning problems: namely 'adversarial' test-time attacks (in several variations) and 'natural' distribution shifts. In this work, we provide a reliable learner with provably optimal guarantees in such settings. We discuss computationally feasible implementations of the learner and further show that our algorithm achieves strong positive performance guarantees on several natural examples: for example, linear separators under log-concave distributions or smooth boundary classifiers under smooth probability distributions.

## 1 Introduction

The question of providing reliability guarantees on the output of learned classifiers has been studied previously in the classical learning setting where the training and test data are independent and identically distributed (i.i.d.) draws from the same distribution [RS88, EYW10, EYW12]. Conceptually, a *reliable* learner outputs a prediction and may output a correctness guarantee. We know that the learner is correct on all points with the guarantee as long as the learning-theoretic assumptions hold, e.g., realizability. While a trivial model that abstains from providing any guarantee is also a reliable learner, we are interested in a reliable learner that provides the guarantee on as many points as possible (*useful* in the sense of Rivest and Sloan [RS88]). [EYW10] provides a characterization of optimal reliable learners in this classical learning setting.

However, the assumption that the training and test data are drawn from the same distribution is often violated in practice. The mismatch may take the form of a 'natural distribution shift' when the test distribution is different from the training distribution or 'adversarial attacks' when there is an adversary that can perturb a test data point with the goal of changing the model prediction. This is frequently accompanied by a significant performance drop, as well as the inability to guarantee the usefulness of the algorithm. As a result, there is a significant interest in the study of test-time attacks [GSS15, CW17, MMS+18] and distribution shift [LWS18, RRSS19, MTR+21] among the applied machine learning community. Furthermore, recently there has been growing interest in the theoretical machine learning community for designing approaches with provable guarantees under test time attacks [AKM19, MHS19, MHS22] as well as renewed interest in distribution shift [BDBCP06, MMR08, HK19]. All the prior theoretical work in the literature has mainly focused on the effect of attacks or distribution shift on average error rate (e.g. [BDBCP06, AKM19]). However,

this neglects a major relevant concern for users of machine learning algorithms, namely the ability to provide correctness guarantees for individual predictions: i.e., reliability.

In this work, we advance this line of work by developing a general understanding of how to learn reliably in the presence of corruptions or changes to the test set, specifically under adversarial test-time attacks as well as distribution shift between the training (source) and test (target) data.

**Our results**. We consider algorithms that provide robustly-reliable predictions which are guaranteed to be correct under standard assumptions from statistical learning theory, for both test-time attacks and distribution shift. Our first main set of results tackles the challenging case of adversarial test-time perturbations. For this setting, we introduce a novel compelling reliability criterion on a learner that particularly captures the challenge of reliability under the test-time attacks. Given a test point $z$, a *robustly-reliable* classifier either abstains from prediction, or outputs both a prediction $y$ and a reliability guarantee $\eta$ with the guarantee that $y$ is correct unless one of two bad events has occurred: 1) the true target function does not belong to the given hypothesis set $\mathcal{H}$ or, 2) a test-point $z$ is perturbed from its original point by adversarial strength of at least $\eta$ (measured in the relevant metric). In the case of distribution shift, we provide novel analysis and a complexity measure that extend the classical notion of reliable learning to the setting when the test distribution is allowed to be an arbitrary new distribution.

## 1.1 Summary of contributions

1. We propose robustly-reliable learners for test-time attacks which guarantee reliable learning in the presence of test-time attacks, and characterize the region of instance space where they are simultaneously robust and reliable. Specifically, under the realizable setting, for adversarial perturbations within metric balls around the test points, we use the radius of the metric ball as a natural notion of adversarial strength. We output a reliability radius $\eta$ with a guarantee that our prediction on a point is correct as long as it was perturbed with a distance less than $\eta$ (under a given metric). We further show that our proposed robustly-reliable learner achieves pointwise optimal values for this reliability radius: that is, no robustly-reliable learner can output a reliability radius larger than our learner for any point in the instance space (Theorem B.1, B.2).
2. The pointwise optimal algorithm is easy to derive from our definition. We discuss a computationally efficient implementation of the optimal learners. (Section 4).
3. We discuss variants of these algorithms and guarantees appropriate for three different variants of adversarial losses studied in the literature: depending on whether the perturbed point must have the same label as the original point, or in lieu of this, whether the algorithm should predict the true label of the perturbed point, or the same label as the original point (Definition 1).
4. We further introduce a safely-reliable region, which captures the challenge caused by the adversary's ability to perturb a test point to cause a reduction in our reliability radius (Definition 6). As examples, we show that the safely-reliable region can be large for linear separators under log-concave distributions and for classifiers with smooth decision boundaries under nearly-uniform distributions and as a consequence, the robustly-reliable region is large as well (Theorem 3.3).
5. We extend this characterization to abstention-based reliable predictions for arbitrary adversarial perturbation sets, where we no longer restrict ourselves to metric balls. We again get a tight characterization of the robustly-reliable region (Theorem C.1).
6. We also consider reliability in the distribution shift setting where the test data points come from a different distribution. We introduce a novel refinement to the notion of disagreement coefficient [Han07], to measure the **transferability of reliability guarantees** across distributions. We provide bounds on the probability mass of the reliable region under transfer for several interesting examples including, when learning linear separators, transfer from $\beta_1$ log-concave to $\beta_2$ log-concave and to $s$-concave distributions (Theorems G.1, G.2). We additionally bound the probability of the reliable region for learning classifiers with general smooth classification boundaries, for transfer between smooth distributions (Theorem G.3).
7. We further extend our reliability results to the setting of robustness tranfer, where the test data is simultaneously under adversarial perturbations as well as distribution shift (Section J).
8. Finally, we demonstrate that it is possible extend our results into the agnostic setting. (Section 7)

**Conceptual advances over prior work.** Prior works on certified robustness [SKL17, CRK19, WLF22] have examined pointwise consistency guarantees. The certified robustness guarantee is only that a prediction does not change with an adversarial perturbation, but it does not guarantee that the

prediction is correct (neither for the original point nor the perturbation); in particular, a constant function is always certified robust but it may not be useful. In contrast, our notion of robustly-reliable learner guarantees that, for any test point $x$ and perturbation $z$, if $z$ has a distance less than $\eta$ to $x$ ($\eta$ = reliability radius), then the prediction will be "correct" (robust loss zero) in a sense informed by which robust loss we are addressing; we discuss this idea for several different losses, leading to different interpretations of this guarantee. In particular, for the stability loss, the prediction being "correct" means that it predicts the true label of the original point $x$; this implies certified robustness, but is even stronger, since it also guarantees the correct label. Prior work [BBHS22] introduces the notion of a robustly-reliable learner for poisoning attacks which is different from our definition that is tailored to test-time attacks with a guarantee in terms of a reliability radius. In distribution shifts setting, we are the first to assess the transferability of reliability guarantees which differ from a widely-studied metric of average error rate. For additional related work, we refer to Appendix A.

## 2 Preliminaries and problem formulation

Let $\mathcal{X}$ denote the instance space and $\mathcal{Y} = \{0, 1\}$ be the label space. Let $\mathcal{H}$ be a hypothesis class. The learner $\mathcal{L}$ is given access to a labeled sample $S = \{(x_i, y_i)\}_{i=1}^m$ drawn from a distribution $\mathcal{D}$ over $\mathcal{X} \times \mathcal{Y}$ and learns a concept $h^{\mathcal{L}} : \mathcal{X} \to \mathcal{Y}$. In the realizable setting, we assume we have a hypothesis (concept) class $\mathcal{H}$ and target concept $h^* \in \mathcal{H}$ such that the *true label* of any $x \in \mathcal{X}$ is given by $h^*(x)$. In particular, $S = \{(x_i, h^*(x_i))\}_{i=1}^m$ in this setting. Given the 0-1 loss function $\ell : \mathcal{H} \times \mathcal{X} \to \{0, 1\}$, define $\mathrm{err}_S(h, \ell) = \frac{1}{m} \sum_{(x,y) \in S} \ell(h, x)$. We use $\mathcal{D}_{\mathcal{X}}$ to denote the marginal distribution over $\mathcal{X}$. We use $\mathbb{I}[\cdot]$ to denote the indicator function that takes values in $\{0, 1\}$. We also define $B_{\mathcal{D}}^{\mathcal{H}}(h^*, r) = \{h \in \mathcal{H} \mid \Pr_{\mathcal{D}}[h(x) \neq h^*(x)] \leq r\}$ as the set of hypotheses in $\mathcal{H}$ that disagree with $h^*$ with probability at most $r$. During test-time, the learner makes a prediction on a test-point $z \in \mathcal{X}$. We consider the following settings

1. **Adversarial test-time attack.** We consider adversarial attacks with perturbation function $\mathcal{U} : \mathcal{X} \to 2^{\mathcal{X}}$ that can perturb a test point $x$ to an arbitrary point $z$ from the perturbation set $\mathcal{U}(x) \subseteq \mathcal{X}$ [MHS19]. We assume that the adversary has access to the learned concept $h^{\mathcal{L}}$ as well as the test point $x$, and can perturb this data point to any $z \in \mathcal{U}(x)$ and then provide this perturbed data point to the learner at test-time. We want to provide pointwise robustness and reliability guarantees in this setting. We will assume that $x \in \mathcal{U}(x)$ for all $x \in \mathcal{X}$. For any point $z$, we have $\mathcal{U}^{-1}(z) := \{x \in \mathcal{X} | z \in \mathcal{U}(x)\}$, the set of points that can be perturbed to $z$. We use *perturbation* to refer to a point $z \in \mathcal{U}(x)$ and the perturbation sets $\mathcal{U}(x)$ interchangeably.
2. **Distribution shift.** We consider when a test point $z$ is drawn from a different distribution from the training samples. In this case, we want to provide a pointwise reliability guarantee. We will discuss more on this in Section 5.

### 2.1 Robust loss functions

In the applied and theoretical literature, various definitions of adversarial success have been explored, each dependent on the interpretation of robustness; depending on whether the perturbed point must have the same label as the original point, or in lieu of this, whether the algorithm should predict the true label of the perturbed point, or the same label as the original point. To capture these, we formally consider the following loss functions.

**Definition 1** (Robust loss functions). *For a hypothesis $h$, a test point $x$, and a perturbation function $\mathcal{U}$, we consider the following adversarially successful events.*

1. *Constrained Adversary loss [SZS+14, BBSZ23]. There exists a perturbation $z$ of $x$ that does not change the true label of an original point $x$ but $h(z)$ is incorrect.*

$$\ell_{CA}^{h^*}(h, x) = \sup_{\substack{z \in \mathcal{U}(x) \\ h^*(z) = h^*(x)}} \mathbb{I}[h(z) \neq h^*(z)].$$

*For a fixed perturbation $z \in \mathcal{U}(x)$, define $\ell_{CA}^{h^*}(h, x, z) = \mathbb{I}[h(z) \neq h^*(z) \wedge h^*(z) = h^*(x)]$.*
2. *True Label loss [ZL19, GKKW21]. There exists a perturbation $z$ of $x$ such that $h(z)$ is incorrect.*

$$\ell_{TL}^{h^*}(h, x) = \sup_{z \in \mathcal{U}(x)} \mathbb{I}[h(z) \neq h^*(z)].$$

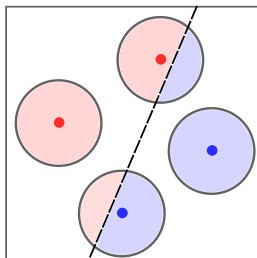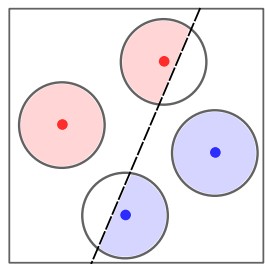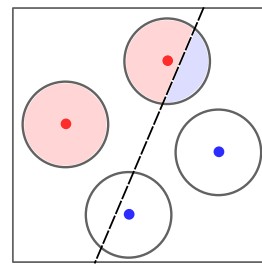

Figure 1: Different perturbation sets considered in $\ell_{\mathrm{TL}}, \ell_{\mathrm{ST}}$ (left), $\ell_{\mathrm{CA}}$ (mid) and $\ell_{\mathrm{IA}}$ (right). The dashed line represents the decision boundary of $h^*$ and the background color of red and blue represents the label 0 and 1 respectively. The ball around each point describes the possible perturbation set $\mathcal{U}(x)$ and the shaded area inside each ball is the allowed perturbation. In $\ell_{\mathrm{TL}}, \ell_{\mathrm{ST}}$, we consider all perturbation in $\mathcal{U}(x)$ while in $\ell_{\mathrm{CA}}$, we consider perturbations that do not change the true label of the perturbed point. Lastly, in $\ell_{\mathrm{IA}}$, an adversary only perturb points where the original true label is 0.

*In this case, if the true label of the $z$ changes, then the learner need to match its prediction with the new label. For a fixed perturbation $z \in \mathcal{U}(x)$, define $\ell_{TL}^{h^*}(h, x, z) = \mathbb{I}[h(z) \neq h^*(z)]$.*

3. **Stability loss** *[AKM19, MHS19, MHS22]. There exists a perturbation $z$ of $x$ such that $h(z)$ is different from $h^*(x)$.*

$$\ell_{ST}^{h^*}(h, x) = \sup_{z \in \mathcal{U}(x)} \mathbb{I}[h(z) \neq h^*(x)].$$

*In this case, we focus on the consistency aspect where we want the prediction of any perturbation $z$ to be the same as the prediction of $x$ and this has to be correct w.r.t. $x$ i.e. equals to $h^*(x)$. For a fixed perturbation $z \in \mathcal{U}(x)$, define $\ell_{ST}^{h^*}(h, x, z) = \mathbb{I}[h(z) \neq h^*(x)]$.*

4. **Incentive-aware Adversary loss** *[ZC21]. We take inspiration from economics application where we assume that the label $1$ is a more favorable outcome e.g. loan approval for which an adversary has no incentive to make any perturbation when the original label is $1$. Define the perturbation set*

$$\mathcal{U}_{IA}(x, h^*) = \begin{cases} \mathcal{U}(x) & ; h^*(x) = 0 \\ \{x\} & ; h^*(x) = 1 \end{cases}$$

*and define an incentive-aware adversary loss as*

$$\ell_{IA}^{h^*}(h, x) = \sup_{z \in \mathcal{U}_{IA}(x, h^*)} \mathbb{I}[h(z) \neq h^*(x)].$$

*For a fixed perturbation $z \in \mathcal{U}(x)$, define $\ell_{IA}^{h^*}(h, x, z) = \mathbb{I}[h(z) \neq h^*(x) \wedge z \in \mathcal{U}_{IA}(x, h^*)]$.*

*We say that $h$ is robust to a perturbation function $\mathcal{U}$ at $x$ w.r.t. a robust loss $\ell$ if $\ell^{h^*}(h, x) = 0$.*

**Remark.** $\ell_{\mathrm{ST}}, \ell_{\mathrm{IA}}$ are robust losses that we can always evaluate in practice on the training data since we are comparing $h(z)$ with $h^*(x)$ which is known to us on the training data. For $\ell_{\mathrm{CA}}, \ell_{\mathrm{TL}}$, we are comparing $h(z)$ with $h^*(z)$ for which $z$ may lie outside of the support of the natural data distribution and we may not have access to $h^*(z)$. We illustrate the relationship between these losses by making a few useful observations.

- In the robustly-realizable case [MHS20] when the perturbation function $\mathcal{U}$ does not change the *true* label of any $x$ in the training or test data, then all the losses $\ell_{\mathrm{CA}}, \ell_{\mathrm{TL}}, \ell_{\mathrm{ST}}$ are equivalent. This corresponds to a common assumption in the adversarial robustness literature, that the perturbations are "human-imperceptible", which is usually quantified as the set of perturbations within a small metric ball around the data point.
- We provide an illustration of the perturbation set considered in various robust losses in Figure 1. By considering these perturbation set, we have the following implication $\ell_{\mathrm{TL}} \to \ell_{\mathrm{CA}}, \ell_{\mathrm{ST}} \to \ell_{\mathrm{CA}}$ and $\ell_{\mathrm{ST}} \to \ell_{\mathrm{IA}}$ where $\ell_1 \to \ell_2$ means robustness w.r.t. $\ell_1$ implies robustness w.r.t. $\ell_2$.

# 3 Robustly-reliable learners w.r.t. metric ball attacks

Although our robust losses are defined for any general perturbation set, we first consider the case where the perturbation sets are balls in some metric space. Such attacks are widely studied in the literature, in particular, for balls with bounded $L_p$-norm. Moreover, the radius of the metric ball serves as a natural notion of adversarial strength that allows us to quantify the level of robustness. We will later (Theorem C.1) present results for general perturbation sets as well.

Let $\mathcal{M} = (\mathcal{X}, d)$ be a metric space equipped with distance metric $d$. We use the notation $\mathbf{B}_{\mathcal{M}}(x, r) = \{x' \in \mathcal{X} \mid d(x, x') \leq r\}$ (resp. $\mathbf{B}_{\mathcal{M}}^o(x, r) = \{x' \in \mathcal{X} \mid d(x, x') < r\}$) to denote a closed (resp. open) ball of radius $r$ centered at $x$. We will sometimes omit the underlying metric $\mathcal{M}$ from the subscript to reduce notational clutter. We formally define a metric ball attack as follows.

**Definition 2.** *__Metric-ball attacks__ are defined as the class of perturbation functions $\mathcal{U}_{\mathcal{M}} = \{u_\eta : \mathcal{X} \to 2^{\mathcal{X}} \mid u_\eta(x) = \mathbf{B}_{\mathcal{M}}(x, \eta)\}$, induced by the metric $\mathcal{M} = (\mathcal{X}, d)$ defined over the instance space.*

At test-time, given a test-point $z \in \mathcal{X}$, we would like to make a prediction at $z$ with a reliability guarantee. We consider this type of learner, a *robustly-reliable* learner defined formally as follows.

**Definition 3** (Robustly-reliable learner w.r.t. $\mathcal{M}$-ball attacks). *A learner $\mathcal{L}$ is robustly-reliable w.r.t. $\mathcal{M}$-ball attacks for hypothesis space $\mathcal{H}$ and robust loss function $\ell$ if, __for any target__ concept $h^* \in \mathcal{H}$, given $S$ labeled by $h^*$, the learner outputs functions $h_S^{\mathcal{L}} : \mathcal{X} \to \mathcal{Y}$ and $r_S^{\mathcal{L}} : \mathcal{X} \to [0, \infty) \cup \{-1\}$ such that for all $x, z \in \mathcal{X}$ if $r_S^{\mathcal{L}}(z) = \eta > 0$ and $z \in \mathbf{B}_{\mathcal{M}}^o(x, \eta)$ then $\ell^{h^*}(h_S^{\mathcal{L}}, x, z) = 0$. Further, if $r_S^{\mathcal{L}}(z) = 0$, then $h^*(z) = h_S^{\mathcal{L}}(z)$.*

Note that $\mathcal{L}$ outputs a prediction and a real value $r$ (the "reliability radius") for any test input. $r = -1$ corresponds to abstention (even in the absence of perturbation) i.e. when the learner is incapable of giving a reliability guarantee for that prediction), and $r = \eta > 0$ is a guarantee from the learner that if the adversary's attack is in $\mathbf{B}_{\mathcal{M}}^o(x, \eta)$ then we are correct i.e. if an adversary changes the original test point $x$ to $z$, the attack will not succeed if the adversarial budget is less than $\eta$. Lastly, when $r = 0$, the learner provides a guarantee that the learner's prediction at $z$ is correct.

**Definition 4** (Robustly-reliable region w.r.t. $\mathcal{M}$-ball attacks). *For a robustly-reliable learner $\mathcal{L}$ w.r.t. $\mathcal{M}$-ball attacks for sample $S$, hypothesis space $\mathcal{H}$ and robust loss function $\ell$ defined above, the robustly-reliable region of $\mathcal{L}$ at a reliability level $\eta$ is defined as $RR^{\mathcal{L}}(S, \eta) = \{x \in \mathcal{X} \mid r_S^{\mathcal{L}}(x) \geq \eta\}$ for sample $S$ and $\eta \geq 0$.*

The robustly-reliable region contains all points with a reliability guarantee of at least $\eta$. We use $\mathrm{RR}_W^{\mathcal{L}}$ to denote robustly-reliable regions with respect to losses $\ell_W$ for $W \in \{\mathrm{CA}, \mathrm{TL}, \mathrm{ST}, \mathrm{IA}\}$. A natural goal is to find a robustly-reliable learner $\mathcal{L}$ that has the largest robustly-reliable region possible. First, we note that predictions that are known by the learner to be correct are still known to be correct even when the test points are attacked. Therefore, a test point $z$ lies in the robustly-reliable region w.r.t. $\ell_{\mathrm{CA}}, \ell_{\mathrm{TL}}$, as long as we can be sure that $h_S^{\mathcal{L}}(z)$ is correct. This is equivalent to $z$ being classified perfectly, i.e. according to the true label. Therefore, the robustly-reliable region w.r.t. $\ell_{\mathrm{CA}}, \ell_{\mathrm{TL}}$ is given by the agreement region of the version space, which is the largest region where we can be sure of what the correct label is in the absence of any adversarial attack [EYW10]. We recall the definition of version space [Mit82] and agreement region [CAL94, BBL06].

**Definition 5.** *For a set $H \subseteq \mathcal{H}$ of hypothesis, and any set of samples $S$, let $\mathrm{DIS}(H) = \{x \in \mathcal{X} : \exists h_1, h_2 \in \mathcal{H} \text{ s.t. } h_1(x) \neq h_2(x)\}$ be the __disagreement region__ and $\mathrm{Agree}(H) = \mathcal{X} \setminus \mathrm{DIS}(H)$ be the __agreement region__. Let $\mathcal{H}_0(S) = \{h \in \mathcal{H} \mid \mathrm{err}_S(h) = 0\}$ be a __version space__: the set of all hypotheses that correctly classify $S$. More generally, $\mathcal{H}_\nu(S) = \{h \in \mathcal{H} \mid \mathrm{err}_S(h) \leq \nu\}$ for $\nu \geq 0$.*

We can also characterize the robustly-reliable region with respect to other robust losses in terms of the agreement region in the following Theorem.

**Theorem 3.1.** *Let $\mathcal{H}$ be any hypothesis class. With respect to $\mathcal{M}$-ball attacks and $\ell_W$, for $\eta \geq 0$,*

*(a) there exists a robustly-reliable learner $\mathcal{L}$ such that $RR_W^{\mathcal{L}}(S, \eta) \supseteq A_W$, and*
*(b) for any robustly-reliable learner $\mathcal{L}$, $RR_W^{\mathcal{L}}(S, \eta) \subseteq A_W$.*

*Specifically, for the robust loss $\ell_W$, the optimal robustly-reliable region $A_W$ are*

| Robust loss $\ell_W$ | Optimal robustly-reliable region $A_W$ |
|---|---|
| $\ell_{\text{CA}}, \ell_{\text{TL}}$ | $\{z \mid z \in \text{Agree}(\mathcal{H}_0(S))\}$ |
| $\ell_{\text{ST}}$ | $\{z \mid \mathbf{B}^o(z, \eta) \subseteq \text{Agree}(\mathcal{H}_0(S)) \land h(z) = h(x), \forall x \in \mathbf{B}^o(z, \eta), \forall h \in \mathcal{H}_0(S)\}$ |
| $\ell_{\text{IA}}$ | $(A_{\text{ST}} \cap \{z \mid h^*(z) = 1\}) \cup \{z \mid z \in \text{Agree}(\mathcal{H}_0(S)) \land h^*(z) = 0\}$ |

*Proof.* (Sketch) We provide the construction of the optimal robustly-reliable learner $\mathcal{L}_{\text{opt}}$ such that $\text{RR}_W^{\mathcal{L}_{\text{opt}}}(S, \eta) \supseteq A_W$ and later show that for any robustly-reliable learner $\mathcal{L}$, we must also have $\text{RR}_W^{\mathcal{L}}(S, \eta) \subseteq A_W$. We start with $\ell_{\text{CA}}, \ell_{\text{TL}}$, consider a learner $\mathcal{L}_{\text{opt}}$ that predicts using an ERM classifier and outputs $\eta = \infty$ for all points in the agreement region of $\mathcal{H}_0(S)$. Any prediction in $\text{Agree}(\mathcal{H}_0(S))$ is reliable because it also agrees with $h^*$ ($h^* \in \mathcal{H}_0(S)$ by realizability). On the other hand, for $z \in \text{DIS}(\mathcal{H}_0(S))$, there exist $h_1, h_2 \in \mathcal{H}_0(S)$ that disagree on $z$. For any learner $\mathcal{L}$, it is not possible to guarantee that $h^{\mathcal{L}}(z)$ is correct as we may have $h^* = h_1$ or $h^* = h_2$.

Now, for $\ell_{\text{ST}}$, consider a learner $\mathcal{L}_{\text{opt}}$ that classifies using an ERM but the reliability radius is now the largest $\eta > 0$ for which $\mathbf{B}^o(z, \eta) \subseteq \text{Agree}(\mathcal{H}_0(S))$ and $h(x) = h(z), \forall x \in \mathbf{B}^o(z, \eta), h \in \mathcal{H}_0(S)$, else $\eta = 0$ if $z \in \text{Agree}(\mathcal{H}_0(S))$, and $-1$ otherwise. The first condition guarantees that $h(x) = h^*(x), \forall x \in \mathbf{B}^o(z, \eta)$. Combined with the second condition we have $h(z) = h(x) = h^*(x), \forall x \in \mathbf{B}^o(z, \eta)$. Thus, $\mathcal{L}_{\text{opt}}$ is a robustly-reliable learner. On the other hand, for a robustly-reliable learner $\mathcal{L}$, consider $z \in \text{RR}_{\text{ST}}^{\mathcal{L}}(S, \eta)$ for $\eta > 0$. We must have $h^{\mathcal{L}}(z) = h^*(x), \forall x \in \mathbf{B}^o(z, \eta)$. Using a similar argument to the case for $\ell_{\text{CA}}, \ell_{\text{TL}}$, we have $z \in \text{Agree}(\mathcal{H}_0(S))$. If there exists $x \in \mathbf{B}^o(z, \eta)$ that $x \notin \text{Agree}(\mathcal{H}_0(S))$, there exists $h_1, h_2 \in \mathcal{H}_0(S)$ that $h^{\mathcal{L}}(z) \neq h_1(x)$ or $h^{\mathcal{L}}(z) \neq h_2(x)$. It is not possible to guarantee that $h^{\mathcal{L}}(z) = h^*(x)$ as we may have $h^* = h_1$ or $h^* = h_2$. Therefore, we must have $\mathbf{B}^o(z, \eta) \subseteq \text{Agree}(\mathcal{H}_0(S))$. Finally, we cannot have $x \in \mathbf{B}^o(z, \eta)$ that $h(z) = h^*(z) \neq h^*(x)$ since this contradict with $h(z) = h^*(x)$. Therefore, we must have $h^*(x) = h^*(z)$. Since we have $x \in \text{Agree}(\mathcal{H}_0(S))$, this implies that $h(x) = h(z)$ for $h \in \mathcal{H}_0(S)$. For $\ell_{\text{IA}}$, the construction is similar to $\ell_{\text{ST}}$. For full proof, we refer to Appendix B. $\qquad \square$

For $\ell_{\text{ST}}$, the learner is able to certify a subset of the agreement region, which satisfies two additional conditions: $h_S^{\mathcal{L}}$ must be correct on all possible points $x$ that could be perturbed to an observed test point $z$, and the true label of $z$ should match the true label of $x$. We denote the second condition of $h(z) = h(x), \forall x \in \mathbf{B}^o(z, \eta), \forall h \in \mathcal{H}_0(S)$ as the **label consistency condition**. For $\ell_{\text{IA}}$, since robustness w.r.t. $\ell_{\text{ST}}$ implies robustness w.r.t. $\ell_{\text{IA}}$, we have $A_{\text{ST}} \subset A_{\text{IA}}$. In addition, with an incentive-aware adversary, whenever $h(z) = 0$, we must have $h^*(x) = 0$ since the adversary does not perturb a data point with the original label 1. Therefore, we can additionally provide a guarantee on $z$ that lies in the agreement region and $h(z) = 0$. We remark that we can identify the term $h^*(z)$ for any point $z \in \text{Agree}(\mathcal{H}_0(S))$ due to the realizability assumption.

We have provided a robustly-reliable learner with the largest possible robustly-reliable region for losses $\ell_{\text{CA}}, \ell_{\text{TL}}, \ell_{\text{ST}}, \ell_{\text{IA}}$. However, we note that the probability mass of the robustly-reliable region may not be a meaningful way to quantify the overall reliability of a learner because a perturbation $z$ may lie outside of the support of the natural data distribution and have zero probability mass. It seems more useful to measure the mass of points $x$ where any perturbation $z$ of $x$ still lies within the robustly-reliable region. We formally define this region as the safely-reliable region.

**Definition 6** (Safely-reliable region w.r.t. $\mathcal{M}$-ball attacks). *Let $\mathcal{L}$ be a robustly-reliable learner w.r.t. $\mathcal{M}$-ball attacks for sample $S$, hypothesis class $\mathcal{H}$ and robust loss function $\ell$. The safely-reliable region of learner $\mathcal{L}$ at reliability levels $\eta_1, \eta_2$ is defined as*

*1. $SR_{CA}^{\mathcal{L}}(S, \eta_1, \eta_2) = \{x \in \mathcal{X} \mid \mathbf{B}_{\mathcal{M}}(x, \eta_1) \cap \{z \mid h^*(z) = h^*(x)\} \subseteq RR_{CA}^{\mathcal{L}}(S, \eta_2)\}$,*

*2. $SR_W^{\mathcal{L}}(S, \eta_1, \eta_2) = \{x \in \mathcal{X} \mid \mathbf{B}_{\mathcal{M}}(x, \eta_1) \subseteq RR_W^{\mathcal{L}}(S, \eta_2)\}$ for $W \in \{TL, ST\}$,*

*3. $SR_{IA}^{\mathcal{L}}(S, \eta_1, \eta_2) = \{x \in \mathcal{X} \mid h^*(x) = 0 \land \mathbf{B}_{\mathcal{M}}(x, \eta_1) \subseteq RR_{IA}^{\mathcal{L}}(S, \eta_2)\} \cup \{x \in \mathcal{X} \mid h^*(x) = 1 \land x \in RR_{IA}^{\mathcal{L}}(S, \eta_2))\}$.*

The safely-reliable region contains any point that retains a reliability radius of at least $\eta_2$ even after being attacked by an adversary with strength $\eta_1$. In the safely-reliable region, we consider a set of potential natural (before attack) points $x$, while in the robustly-reliable region, we consider a set of potential test points $z$. In the following subsections, we show that in interesting cases commonly studied in the literature, the probability mass of the safely-reliable region is actually quite large.

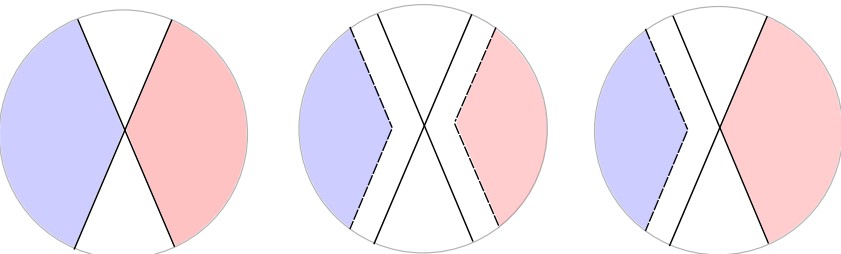

Figure 2: The robustly-reliable region for $\ell_{CA}, \ell_{TL}$ (left), $\ell_{ST}$ (mid) and $\ell_{IA}$(right) for linear separators with an $L_2$-ball perturbation. The background color of blue and red represents the agreement region of class 1 and 0 respectively. In this case, we can remove the label consistency condition and reduce the robustly-reliable region into the '$\eta$-buffered' agreement region.

## 3.1 Safely-reliable region for linear separators under log-concave distributions is large

We provide the probability mass of safely-reliable regions with respect to different losses for linear separators when the data distribution follows an isotropic (mean zero and identity covariance matrix) log-concave (logarithm of density function is concave) distribution under a bounded $L_2$-norm ball attack. For full proof, we refer to Appendix D. We will rely on the following key lemma which states that the agreement region of a linear separator cannot contain points that are arbitrarily close to the decision boundary of $h^*$ for any sample $S$.

**Lemma 3.2.** *Let $\mathcal{D}$ be a distribution over $\mathbb{R}^d$ and $\mathcal{H} = \{h : x \to \text{sign}(\langle w_h, x \rangle) \mid w_h \in \mathbb{R}^d, \|w_h\|_2 = 1\}$ be a class of linear separators. For $h^* \in \mathcal{H}$, for a set of samples $S \sim \mathcal{D}^m$ such that there is no data point in $S$ that lies on the decision boundary, for any $0 < c < d$, there exists $\delta(S, c, d) > 0$ such that for any $x$ with $c \le \|x\| \le d$ and $|\langle w_{h^*}, x \rangle| < \delta$, we have $x \notin \text{Agree}(\mathcal{H}_0(S))$.*

A direct implication of the lemma is that any $L_2$-ball $\mathbf{B}(x, \eta)$ that lies in the agreement region must not contain the decision boundary of $h^*$ and must contain points with the same label. This allows us to remove the *label consistency condition* and instead focus on whether the ball $\mathbf{B}(x, \eta)$ lies in the agreement region. Intuitively, the reliable region is now given by the '$\eta$-buffered' agreement region where we only select points that have a distance at least $\eta$ from the boundary of the agreement region (Figure 2). We provide bounds for the probability mass of the safely-reliable region below and we refer to the full proof in Appendix D.

**Theorem 3.3.** *Let $\mathcal{D}$ be isotropic log-concave over $\mathbb{R}^d$ and $\mathcal{H} = \{h : x \to \text{sign}(\langle w_h, x \rangle) \mid w_h \in \mathbb{R}^d, \|w_h\|_2 = 1\}$ be the class of linear separators. Let $\mathbf{B}(\cdot, \eta)$ be a $L_2$ ball perturbation with radius $\eta$. For $S \sim \mathcal{D}^m$, for $m = \mathcal{O}(\frac{1}{\varepsilon^2}(\text{VCdim}(\mathcal{H}) + \ln \frac{1}{\delta}))$, for an optimal robustly-reliable learner $\mathcal{L}$,*

*(a) $\text{Pr}(SR_{TL}^{\mathcal{L}}(S, \eta_1, \eta_2)) \ge 1 - 2\eta_1 - \tilde{\mathcal{O}}(\sqrt{d}\varepsilon)$ with probability at least $1 - \delta$,*
*(b) $SR_{CA}^{\mathcal{L}}(S, \eta_1, \eta_2) = SR_{TL}^{\mathcal{L}}(S, \eta_1, \eta_2)$ almost surely,*
*(c) $\text{Pr}(SR_{ST}^{\mathcal{L}}(S, \eta_1, \eta_2)) \ge 1 - 2(\eta_1 + \eta_2) - \tilde{\mathcal{O}}(\sqrt{d}\varepsilon)$ with probability at least $1 - \delta$,*
*(d) $\text{Pr}(SR_{IA}^{\mathcal{L}}(S, \eta_1, \eta_2)) \ge 1 - (\eta_1 + \eta_2) - \tilde{\mathcal{O}}(\sqrt{d}\varepsilon)$ with probability at least $1 - \delta$.*

*The $\tilde{\mathcal{O}}$-notation suppresses dependence on logarithmic factors and distribution-specific constants.*

We remark that we can't always remove the label consistency condition for a general perturbation set. For example, consider $\mathcal{U}(x) = \mathbf{B}(x - a, \eta) \cup \{x\} \cup \mathbf{B}(x + a, \eta)$, is made of two $L_2$ balls with center $x - a, x + a$, with appropriate value of $a, \eta$, we may have each ball lie in the different side of the agreement region so that the whole perturbation set lie in the agreement region but contain points with different labels (Figure 3). We also provide bounds on the probability mass of the safely-reliable region for more general concept spaces beyond linear separators, specifically, classifiers with smooth boundaries in Appendix E.

## 4 On computational efficiency

Given the definition, it is possible to implement computationally efficient robustly-reliable learners. For example, for linear separator concept classes under bounded $L_2$-norm attack. The optimal

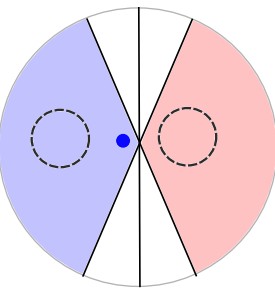

Figure 3: The perturbation set is represented by two dashed balls. This lies inside the agreement region but contains points with different labels.

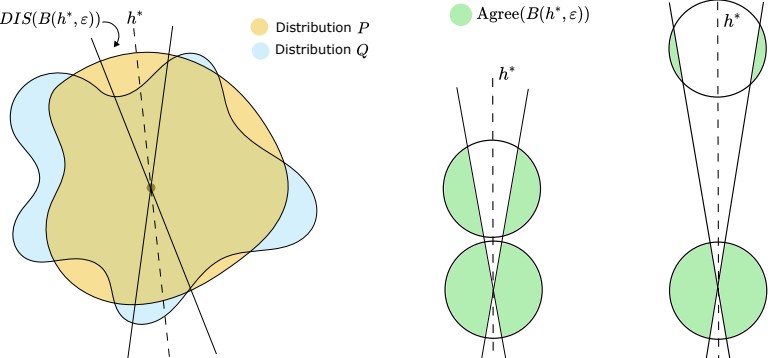

Figure 4: The disagreement region and the agreement region under a distribution shift where $\mathcal{P}$ and $\mathcal{Q}$ are isotropic (left) and where there is a mean shift (right).

robustly-reliable learner $\mathcal{L}_{\text{opt}}$, described above may be implemented via a linearly constrained quadratic program that computes the (squared) distance of the test point $z$ to the closest point $z'$ in the disagreement region. This gives us the reliability radius, since for linear separators one must cross the decision boundary to perturb a point to a differently labeled point

$$\min_{w,w',z'} ||z-z'||^2$$
$$\text{s.t.} \quad \langle w, x_i \rangle y_i \geq 0, \quad \text{for each } (x_i, y_i) \in S,$$
$$\langle w', x_i \rangle y_i \geq 0, \quad \text{for each } (x_i, y_i) \in S,$$
$$\langle w, z' \rangle \langle w', z' \rangle \leq 0.$$

Given training sample $S$, for any given test point $z$, the learner $\mathcal{L}$ can efficiently compute the solution $s^*$ to the above program and output $\sqrt{s^*}$ as the reliability radius. We show that the variant of this objective also provides a reliability radius for a wide range of hypothesis classes under $L_2$ ball attacks (see Lemma F.1). In addition, we can relax this objective into a regularized objective that gives a lower bound on the reliability radius of $||z - z^*||^2$, when $z^*$ is the solution of

$$h_1, h_2, z^* = \underset{h,h',z'}{\operatorname{argmin}} ||z-z'||^2 + \lambda(\hat{R}(h, S \cup \{(z', 0)\}) + \hat{R}(h', S \cup \{(z', 1)\}))$$

when $\hat{R}(h, S)$ is the empirical risk of $h$ on $S$. We provide a more detailed discussion in Appendix F.

## 5   Robustly-reliable learning under distribution shift

We now consider the reliability aspect for distribution shift, a different kind of test-time robustness challenge when the test data comes from a different distribution than the training data. Formally, let $\mathcal{P}$ be the training distribution and let $\mathcal{Q}$ be the test distribution. We assume the *realizable distribution shift* setting, i.e. there is a target concept $h^* \in \mathcal{H}$ such that the true label of any $x \in \mathcal{X}$ is given by $h^*(x)$ at training time and test time, or $\text{err}_{\mathcal{P}}(h^*) = \text{err}_{\mathcal{Q}}(h^*) = 0$. As observed earlier, points that

are known by the learner to be correct (reliable) are still known to be correct even when it is drawn from a different distribution. This reliability guarantee holds even when the distributions $\mathcal{P}$ and $\mathcal{Q}$ do not share a common support, a setting for which many prior theoretical works result in vacuous bounds. For example, suppose $\mathcal{X} = \mathbb{R}^n$, $\mathcal{P}$ and $\mathcal{Q}$ are supported on disjoint $n$-balls, and $\mathcal{H}$ is the class of linear separators. Then the total variation distance, the $\mathcal{H}$-divergence [KBDG04, BDBC+10] as well as the discrepancy distance [MMR08] between $\mathcal{P}$ and $\mathcal{Q}$ are all 1. While recent work of [HK19] does apply in this setting, they do not focus on the reliability guarantee. In this work, we are interested in quantifying the transferability of reliability guarantee transfer between distributions $\mathcal{P}$ and $\mathcal{Q}$. We recall the notion of reliable prediction [EYW10].

**Definition 7** (Reliability). *A learner $\mathcal{L}$ is reliable w.r.t. concept space $\mathcal{H}$ if, for any target concept $h^* \in \mathcal{H}$, given any sample $S$ labeled by $h^*$, the learner outputs functions $h_S^{\mathcal{L}} : \mathcal{X} \to \mathcal{Y}$ and $a_S^{\mathcal{L}} : \mathcal{X} \to \{0, 1\}$ such that for all $x \in \mathcal{X}$ if $a_S^{\mathcal{L}}(x) = 1$ then $h_S^{\mathcal{L}}(x) = h^*(x)$. Else, if $a_S^{\mathcal{L}}(x) = 0$, the learner abstains from prediction. The reliable region of $\mathcal{L}$ is $R^{\mathcal{L}}(S) = \{x \in \mathcal{X} \mid a_S^{\mathcal{L}}(x) = 1\}$.*

We define the following metric to measure the reliability of a learner under distribution shift.

**Definition 8** ($\mathcal{P} \to \mathcal{Q}$ reliable correctness). *The $\mathcal{P} \to \mathcal{Q}$-reliable correctness of $\mathcal{L}$ (at sample rate $m$, for distribution shift from $\mathcal{P}$ to $\mathcal{Q}$) is defined as the expected probability mass of its reliable region under $Q$ when trained on a random training $S \sim \mathcal{P}^m$, i.e. $\Pr_{x \sim Q, S \sim P^m}[x \in R^{\mathcal{L}}(S)]$.*

The disagreement coefficient was originally introduced to study the label complexity in agnostic active learning [Han07] and is also known to characterize reliable learning in the absence of any distribution shift [EYW10]. We propose the following refinement to the notion of disagreement coefficient, which we will use to give bounds on a learner's $\mathcal{P} \to \mathcal{Q}$ reliable correctness.

**Definition 9** ($\mathcal{P} \to \mathcal{Q}$ disagreement coefficient). *For a hypothesis class $\mathcal{H}$, the $\mathcal{P} \to \mathcal{Q}$ disagreement coefficient of $h^* \in \mathcal{H}$ with respect to $\mathcal{H}$ is given by*

$$\Theta_{\mathcal{P} \to \mathcal{Q}}(\varepsilon) = \sup_{r \geq \varepsilon} \frac{\Pr_{\mathcal{Q}}[\mathrm{DIS}(B_{\mathcal{P}}(h^*, r))]}{r},$$

*where $B_{\mathcal{P}}(h^*, r) = \{h \in \mathcal{H} \mid \Pr_{\mathcal{P}}[h(x) \neq h^*(x)] \leq r\}$.*

This quantifies the rate of disagreement over $\mathcal{Q}$ among classifiers which are within disagreement-balls w.r.t. $h^*$ under $\mathcal{P}$, relative to the version space radius. The proposed metric is asymmetric between $\mathcal{P}$ and $\mathcal{Q}$, and also depends on the target concept $h^*$. More simple examples are in Appendix H. We show that the $\mathcal{P} \to \mathcal{Q}$-reliable correctness of our learner may be bounded in terms of the $\mathcal{P} \to \mathcal{Q}$ disagreement coefficient using a uniform convergence based argument. The proof details are in Appendix I.

**Theorem 5.1.** *Let $\mathcal{Q}$ be a realizable distribution shift of $\mathcal{P}$ with respect to $\mathcal{H}$, and $h^* \in \mathcal{H}$ be the target concept. Given sufficiently large sample size $m \geq \frac{c}{\varepsilon^2}(d + \ln \frac{1}{\delta})$, the $\mathcal{P} \to \mathcal{Q}$-reliable correctness of $\mathcal{L}$, the optimal robustly-reliable learner, is at least*

$$\Pr_{x \sim Q, S \sim P^m}[x \in R^{\mathcal{L}}(S)] \geq 1 - \Theta_{\mathcal{P} \to \mathcal{Q}} \cdot \varepsilon - \delta.$$

*Here $c$ is an absolute constant, and $d$ is the VC-dimension of $\mathcal{H}$.*

In Appendix J, we show that this $\mathcal{P} \to \mathcal{Q}$ disagreement coefficient can be small for several examples which implies that it is possible to transfer the reliability guarantee from one distribution to the other. In particular, when learning linear separators, we provide bounds for transferring from $\beta_1$ log-concave to $\beta_2$ log-concave and to $s$-concave distributions (Theorems G.1, G.2). In addition, when learning classifiers with general smooth classification boundaries, we provide bounds for transferring between smooth distributions (Theorem G.3).

# 6 Safely-reliable correctness under distribution shift

There is a growing practical [SSZ+20, SIE+20] as well as recent theoretical interest [DGH+23] in the setting of 'robustness transfer', where one simultaneously expects adversarial test-time attacks as well as distribution shift. We will study the reliability aspect for this more challenging setting. We note that the definition of a robustly-reliable learner does not depend on the data distribution (see

Definition 3) as the guarantee is pointwise. Our optimality result in Section 3 applies even when a test point is drawn from a different distribution $\mathcal{Q}$. In this case, the safely-reliable region instead would have a different probability mass.

**Definition 10** ($\mathcal{P}{\rightarrow}\mathcal{Q}$ safely-reliable correctness). *The $\mathcal{P}{\rightarrow}\mathcal{Q}$ safely-reliable correctness of $\mathcal{L}$ (at sample rate $m$, for distribution shift from $\mathcal{P}$ to $\mathcal{Q}$, w.r.t. robust loss $\ell$) is defined as the probability mass of its safely-reliable region under $Q$, on a sample $S \sim \mathcal{P}^m$, i.e.*

$$PQR_\ell^\mathcal{L}(S, \eta_1, \eta_2) := \Pr_{x \sim Q, S \sim P^m}[x \in SR_\ell^\mathcal{L}(S, \eta_1, \eta_2)].$$

We consider an example when the training distribution $\mathcal{P}$ is isotropic log-concave and the test distribution $\mathcal{Q}_\mu$ is log-concave with its mean shifted by $\mu$ but the covariance matrix is still an identity matrix (see Figure 4, right). We provide the bound on the $\mathcal{P}{\rightarrow}\mathcal{Q}$ safely-reliable correctness of this example in Appendix J (see Theorem J.2).

## 7 Reliability in the agnostic setting

In the above, we have assumed that the training samples $S$ are realizable under our concept class $\mathcal{H}$, i.e. there is a target concept $h^*$ consistent with our (uncorrupted) data. In the agnostic setting, we can have $\min_{h \in \mathcal{H}} \mathrm{err}_S(h) > 0$, meaning no single concept is always correct. We define a $\nu$-*tolerably robustly-reliable* learner under test-time attacks in the agnostic setting as the learner whose reliable predictions agree with every low-error hypothesis (error at most $\nu$) on the training sample ([BBHS22] have proposed the corresponding definition for data poisoning attacks).

**Definition 11** ($\nu$-tolerably robustly-reliable learner w.r.t. $\mathcal{M}$-ball attacks). *A learner $\mathcal{L}$ is robustly-reliable w.r.t. $\mathcal{M}$-ball attacks for sample $S$, hypothesis space $\mathcal{H}$ and robust loss function $\ell$ if, for **every** concept $h^* \in \mathcal{H}$ with $\mathrm{err}_S(h^*) \leq \nu$, the learner outputs functions $h_S^\mathcal{L} : \mathcal{X} \to \mathcal{Y}$ and $r_S^\mathcal{L} : \mathcal{X} \to [0, \infty) \cup \{-1\}$ such that for all $x, z \in \mathcal{X}$ if $r_S^\mathcal{L}(z) = \eta > 0$ and $z \in \mathbf{B}_\mathcal{M}^o(x, \eta)$ then $\ell^{h^*}(h_S^\mathcal{L}, x, z) = 0$. Further, if $r_S^\mathcal{L}(z) = 0$, then $h^*(z) = h_S^\mathcal{L}(z)$. Given sample $S$ such that some concept $h^* \in \mathcal{H}$ satisfies $\mathrm{err}_S(h^*) \leq \nu$, the robustly-reliable region of $\mathcal{L}$ is defined as $RR^\mathcal{L}(S, \nu, \eta) = \{x \in \mathcal{X} \mid r_S^\mathcal{L}(x) \geq \eta\}$ for $\nu, \eta \geq 0$.*

We generalize our results from Section 3 to the agnostic setting (proof details in Appendix K). Here $\mathcal{H}_\nu(S) = \{h \in \mathcal{H} \mid \mathrm{err}_S(h) \leq \nu\}$.

**Theorem 7.1.** *Let $\mathcal{H}$ be any hypothesis class. With respect to $\mathcal{M}$-ball attacks and $\ell_{CA}$, for $\eta \geq 0$,*

(a) *There exists a robustly-reliable learner $\mathcal{L}$ such that $RR_{CA}^\mathcal{L}(S, \nu, \eta) \supseteq \mathrm{Agree}(\mathcal{H}_\nu(S))$,*

(b) *For any robustly-reliable learner $\mathcal{L}$, $RR_{CA}^\mathcal{L}(S, \nu, \eta) \subseteq \mathrm{Agree}(\mathcal{H}_\nu(S))$.*

## 8 Discussion

In this work, we generalize the classical line of works on reliable learning to address challenging test-time environments. We propose a novel robustly-reliability criterion that is applicable to several variations of test-time attacks. Our analysis leads to an easy-to-derive algorithm that can be implemented efficiently in many cases. Additionally, we introduce a $\mathcal{P} \to \mathcal{Q}$ disagreement coefficient to capture the transferability of the reliability guarantee between distributions. The proposed robustly-reliability criterion and the $\mathcal{P} \to \mathcal{Q}$ disagreement coefficient together provide a comprehensive framework for handling test-time attacks and evaluating the reliability of learning models. This contributes to the advancement of reliable learning methodologies in the face of challenging real-world scenarios, facilitating the development of more resilient and trustworthy machine learning systems. Notably, key questions remain open, including, how to efficiently implement the algorithm for a class of neural networks, and how to learn reliably with respect to any general robust loss function?

## 9 Acknowledgements

This work was supported in part by NSF grants CCF-1910321 and SES-1919453, the Defense Advanced Research Projects Agency under cooperative agreement HR00112020003, a Bloomberg Data Science PhD fellowship, and a Simons Investigator Award.

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
