# A  Additional related work

**Reliability.**  A learning model that outputs a confidence level is valuable in practical applications, as it allows us to determine when to trust the model and when to defer the task to a human. However, it is well-known that models like neural networks can exhibit high confidence, yet still produce incorrect results [GPSW17].  To tackle this issue, there has been a line of works on learning algorithms with uncertainty estimate [WR06, BCKW15, GG16, LPB17, MIG$^+$19].  Unlike prior work, our results take into account the relevant notion of robust loss. In particular, we extend the reliability guarantees in perfect selective classification [EYW12] and reliable-useful learning model [RS88] to different robust losses under a test-time attack. Prior work on reliability under data poisoning attacks [BBHS22] obtained similar results on training-time attacks, by providing guarantees that the learner is always correct at any point that it makes a prediction provided the training data corruption does not exceed a point-specific threshold. Our work is also related to learning algorithms with an abstention option [YCJ16, CDM16, CDG$^+$18, PZ21, ZN22].

**Robustness.**  Robustness against adversarial attacks is essential for the safe deployment of machine learning models in the real world.  Our focus in this work is on perturbation attacks, where we aim to provide learners that remain robust even when the test data points are perturbed.  It is known that many modern approaches such as deep neural networks fail in this case even when the perturbation is human-imperceptible [SZS$^+$14, GSS15].  There has been a lot of empirical effort [MMS$^+$18, ZYJ$^+$19, SNG$^+$19, RWK20, TKP$^+$18, CRS$^+$19] as well as theoretical effort [TSE$^+$19, SST$^+$18, JSH20, RXY$^+$20, MHS19, MHS21, GKKM20, BPRZ23] to develop learners with improved robustness, and more broadly to understand various aspects of adversarial robustness. In particular, there is a line of work on certified robustness [CRK19, LAG$^+$19, LCWC19] which provides a pointwise guarantee that the prediction does not change, so long as the attack strength is within a learner-specified 'radius' for the point. While the certified robustness research focuses on this consistency aspect, our work addresses the reliability aspect where we hope to guarantee that the prediction is also correct.

**Distribution shift.**  A distribution shift refers to the phenomenon where the training distribution differs from the test distribution which often leads to a degradation in the learner's performance. This has been studied under several different settings [QCSSL08], ranging from covariate shift [Shi00, HGB$^+$06, BBS07], and domain adaptation [MMR08, BDBC$^+$10, ZLLJ19, ZLWJ20] to transfer learning [PY10, TS10, HK19]. Many algorithms have been proposed to deal with the shift which involves encouraging invariance between different domains [SS16, ABGLP19, RRR21] or taking into account the worst-case subpopulation [ND16, DN21, SKHL20, CWG$^+$19]. While prior work typically focuses on the average performance on the target domain or subpopulation, we provide point-wise reliability guarantees.

**Learning with noise.**  There is extensive classic literature on learning methods which are tolerant or robust to noise [KV94, Vap98]—including efficient learning under bounded or Massart noise [ABHU15, DGT19], agnostic active learning [KKMS08, Dan16], learning under malicious noise [ABL14, BH21], to list a few.  Recent work has considered reliable learning under some of these classic noise models [HKLM20, BBHS22].

# B  Additional proof details for robustly-reliable learners w.r.t. metric ball attacks

**Theorem B.1.** *Let $\mathcal{H}$ be any hypothesis class. With respect to $\mathcal{M}$-ball attacks and $\ell_{CA}$, for $\eta \geq 0$,*

*(a) there exists a robustly-reliable learner $\mathcal{L}$ such that $RR_{CA}^{\mathcal{L}}(S, \eta) \supseteq \mathrm{Agree}(\mathcal{H}_0(S))$, and*
*(b) for any robustly-reliable learner $\mathcal{L}$, $RR_{CA}^{\mathcal{L}}(S, \eta) \subseteq \mathrm{Agree}(\mathcal{H}_0(S))$.*

*The results hold for $RR_{TL}^{\mathcal{L}}$ as well.*

*Proof.* (Proof of Theorem B.1) The robustly-reliable learner $\mathcal{L}$ is given as follows. Set $h_S^{\mathcal{L}} = \mathrm{argmin}_{h \in \mathcal{H}} \mathrm{err}_S(h)$ i.e. an ERM over $S$, and $r_S^{\mathcal{L}}(z) = \infty$ if $z \in \mathrm{Agree}(\mathcal{H}_0(S))$, else $r_S^{\mathcal{L}}(z) = -1$. By realizability, $\mathrm{err}_S(h_S^{\mathcal{L}}) \leq \mathrm{err}_S(h^*) = 0$, or $h_S^{\mathcal{L}} \in \mathcal{H}_0(S)$. We first show that $\mathcal{L}$ is robustly-reliable. For $z \in \mathcal{X}$, if $r_S^{\mathcal{L}}(z) = \eta > 0$, then $z \in \mathrm{Agree}(\mathcal{H}_0(S))$. We have $h^*(z) = h_S^{\mathcal{L}}(z)$ since the classifiers

$h^*, h_S^{\mathcal{L}} \in \mathcal{H}_0(S)$ and $z$ lies in the agreement region of classifiers in $\mathcal{H}_0(S)$ in this case. Thus, we have $\ell_{\mathrm{CA}}^{h^*}(h_S^{\mathcal{L}}, x, z) = 0$ for any $x$ such that $z \in \mathbf{B}_{\mathcal{M}}^o(x, \eta)$. The $\eta = 0$ case corresponds to reliability in the absence of test-time attack, so [EYW10] applies. Therefore, $\mathrm{RR}_{\mathrm{CA}}^{\mathcal{L}}(S, \eta) \supseteq \mathrm{Agree}(\mathcal{H}_0(S))$ for all $\eta \geq 0$ follows from the setting $r_S^{\mathcal{L}}(z) = \infty$ if $z \in \mathrm{Agree}(\mathcal{H}_0(S))$.

Conversely, let $z \in \mathrm{DIS}(\mathcal{H}_0(S))$. There exist $h_1, h_2 \in \mathcal{H}_0(S)$ such that $h_1(z) \neq h_2(z)$. If possible, let there be a robustly-reliable learner $\mathcal{L}$ such that $z \in \mathrm{RR}_{\mathrm{CA}}^{\mathcal{L}}(S, \eta)$ for some $\eta > 0$. By definition of the robust-reliability region, we must have $r_S^{\mathcal{L}}(z) > 0$. By definition of a ball, we have $z \in \mathbf{B}_{\mathcal{M}}^o(z, \eta)$ for any $\eta > 0$, and therefore $\ell_{\mathrm{CA}}^{h^*}(h_S^{\mathcal{L}}, z, z) = 0$. But then we must have $h_S^{\mathcal{L}}(z) = h^*(z)$ by definition of $\ell_{\mathrm{CA}}$. But we can set $h^* = h_1$ or $h^* = h_2$ since both are consistent with $S$. But $h_1(z) \neq h_2(z)$, and therefore $h_S^{\mathcal{L}}(z) \neq h^*(z)$ for one of the above choices for $h^*$, contradicting that $\mathcal{L}$ is robustly-reliable. $\qquad\square$

**Theorem B.2.** *Let $\mathcal{H}$ be any hypothesis class. With respect to $\mathcal{M}$-ball attacks and $\ell_{ST}$, for $\eta \geq 0$,*

*(a) there exists a robustly reliable learner $\mathcal{L}$ such that $RR_{ST}^{\mathcal{L}}(S, \eta) \supseteq A_{ST}$, and*
*(b) for any robustly-reliable learner $\mathcal{L}$, $RR_{ST}^{\mathcal{L}}(S, \eta) \subseteq A_{ST}$,*

*where $A_{ST} = \{z \mid \mathbf{B}^o(z, \eta) \subseteq \mathrm{Agree}(\mathcal{H}_0(S)) \wedge \forall h \in \mathcal{H}_0(S), h(x) = h(z), \forall x \in \mathbf{B}^o(z, \eta)\}$.*

*Proof.* (Proof of Theorem B.2) Given sample $S$, consider the learner $\mathcal{L}$ which outputs $h_S^{\mathcal{L}} = \mathrm{argmin}_{h \in \mathcal{H}} \mathrm{err}_S(h)$, and $r_S^{\mathcal{L}}(z)$ is given by the largest $\eta > 0$ for which $\mathbf{B}^o(z, \eta) \subseteq \mathrm{Agree}(\mathcal{H}_0(S))$ and $h(x) = h(z), \forall x \in \mathbf{B}^o(z, \eta), h \in \mathcal{H}_0(S)$, else $\eta = 0$ if $z \in \mathrm{Agree}(\mathcal{H}_0(S))$, and $-1$ otherwise. Note that the supremum exists here since a union of open sets is also open. By realizability, $\mathrm{err}_S(h_S^{\mathcal{L}}) \leq \mathrm{err}_S(h^*) = 0$, or $h_S^{\mathcal{L}} \in \mathcal{H}_0(S)$. We first show that $\mathcal{L}$ is robustly-reliable w.r.t. $\mathcal{M}$ for loss $\ell_{\mathrm{ST}}$. For $z \in \mathcal{X}$, if $r_S^{\mathcal{L}}(z) = \eta \geq 0$, then $\mathbf{B}^o(z, \eta) \subseteq \mathrm{Agree}(\mathcal{H}_0(S))$, in particular $z \in \mathrm{Agree}(\mathcal{H}_0(S))$. Moreover, by definition, for any $x \in \mathbf{B}^o(z, \eta)$, we have $h_S^{\mathcal{L}}(z) = h_S^{\mathcal{L}}(x)$ by construction. Putting together, and using the property that distance functions of a metric are symmetric, we have $h_S^{\mathcal{L}}(z) = h^*(x)$ for any $x$ such that $z \in \mathbf{B}^o(x, \eta)$. Thus, we have $\ell_{\mathrm{ST}}^{h^*}(h_S^{\mathcal{L}}, x, z) = 0$ for any $x$ such that $z \in \mathbf{B}^o(x, \eta)$. Thus $\mathcal{L}$ satisfies Definition 3.

Conversely, we will show that for any robustly-reliable learner $\mathcal{L}$ w.r.t. $\ell_{\mathrm{ST}}$, for any $\eta > 0$,

$$\mathrm{RR}_{\mathrm{ST}}^{\mathcal{L}}(S, \eta) \subseteq \mathrm{Agree}(\mathcal{H}_0(S)),$$

which follows from similar arguments from Theorem B.1 which also apply to the $\ell_{\mathrm{ST}}$ loss. Let $z \in \mathrm{DIS}(H_0(S))$. There exists $h_1, h_2 \in H_0(S)$ such that $h_1(z) \neq h_2(z)$. If possible, let there be a robustly-reliable learner $\mathcal{L}$ such that $z \in \mathrm{RR}_{\mathrm{ST}}^{\mathcal{L}}(S, \eta)$ for some $\eta > 0$. By definition of the robust-reliability region, we must have $r_S^{\mathcal{L}}(z) > 0$. By definition of a closed ball, we have $z \in \mathbf{B}_{\mathcal{M}}^o(z, \eta)$ for any $\eta > 0$, and therefore $\ell_{\mathrm{ST}}^{h^*}(h_S^{\mathcal{L}}, z, z) = \mathbb{I}[h_S^{\mathcal{L}}(z) \neq h^*(z)] = 0$ which implies that $h_S^{\mathcal{L}}(z) = h^*(z)$. But we can set $h^* = h_1$ or $h^* = h_2$ since both are consistent with $S$. But $h_1(z) \neq h_2(z)$, and therefore $h_S^{\mathcal{L}}(z) \neq h^*(z)$ for one of the above choices for $h^*$, contradicting that $\mathcal{L}$ is robustly-reliable. Next, we will show that, for any $\eta > 0$,

$$\mathrm{RR}_{\mathrm{ST}}^{\mathcal{L}}(S, \eta) \subseteq \{z \mid \mathbf{B}_{\mathcal{M}}^o(z, \eta) \subseteq \mathrm{Agree}(\mathcal{H}_0(S))\}.$$

We will prove this by contradiction. Suppose $z \in \mathrm{Agree}(H_0(S))$, but there exists $x' \in \mathbf{B}_{\mathcal{M}}^o(z, \eta)$ such that $x' \notin \mathrm{Agree}(H_0(S))$. Let there be a robustly-reliable learner $\mathcal{L}$ such that $z \in \mathrm{RR}_{\mathrm{ST}}^{\mathcal{L}}(S, \eta)$. By definition, we have $\ell_{\mathrm{ST}}^{h^*}(h_S^{\mathcal{L}}, x, z) = 0$ for any $x$ that $z \in \mathbf{B}_{\mathcal{M}}^o(x, \eta)$. This implies that $\ell_{\mathrm{ST}}^{h^*}(h_S^{\mathcal{L}}, x', z) = 0$ that is $h_S^{\mathcal{L}}(z) = h^*(x')$. Because $x' \notin \mathrm{Agree}(H_0(S))$, there exists $h_1, h_2 \in H_0(S)$ such that $h_1(x') \neq h_2(x')$. We can set $h^* = h_1$ or $h^* = h_2$ since both are consistent with $S$. But $h_1(x') \neq h_2(x')$, and therefore $h_S^{\mathcal{L}}(z) \neq h^*(z)$ for one of the above choices for $h^*$, contradicting that $\mathcal{L}$ is robustly-reliable. Finally, we will show that, for any $\eta > 0$,

$$\mathrm{RR}_{\mathrm{ST}}^{\mathcal{L}}(S, \eta) \subseteq \{z \mid \mathbf{B}_{\mathcal{M}}^o(z, \eta) \subseteq \mathrm{Agree}(\mathcal{H}_0(S)) \wedge h(x) = h(z), \forall x \in \mathbf{B}_{\mathcal{M}}^o(z, \eta), h \in \mathcal{H}_0(S)\}.$$

Let $z$ be a data point such that $\mathbf{B}_{\mathcal{M}}^o(z, \eta) \subseteq \mathrm{Agree}(H_0(S))$ but there exists $x' \in \mathbf{B}_{\mathcal{M}}^o(z, \eta)$ such that $h(x') \neq h(z)$ for $h \in H_0(S)$. Let there be a robustly-reliable learner such that $z \in \mathrm{RR}_{\mathrm{ST}}^{\mathcal{L}}(S, \eta)$. This implies that $\ell_{\mathrm{ST}}(h_S^{\mathcal{L}}, x, z) = 0$ for any $x$ that $z \in \mathbf{B}_{\mathcal{M}}^o(z, \eta)$. However, $\ell_{\mathrm{ST}}(h_S^{\mathcal{L}}, x', z) = \mathbb{I}[h_S^{\mathcal{L}}(z) \neq h^*(x')] = \mathbb{I}[h_S^{\mathcal{L}}(z) \neq h_S^{\mathcal{L}}(x')] \neq 0$, contradicting that $\mathcal{L}$ is robustly-reliable. $\qquad\square$

**Theorem B.3.** *Let $\mathcal{H}$ be any hypothesis class. With respect to $\mathcal{M}$-ball attacks and $\ell_{IA}$, for $\eta \geq 0$,*

*(a) there exists a robustly reliable learner $\mathcal{L}$ such that $RR_{IA}^{\mathcal{L}}(S, \eta) \supseteq A_{IA}$, and*
*(b) for any robustly-reliable learner $\mathcal{L}$, $RR_{IA}^{\mathcal{L}}(S, \eta) \subseteq A_{IA}$,*

*where $A_{IA} = (A_{ST} \cap \{z \mid h^*(z) = 1\}) \cup \{z \mid z \in \text{Agree}(\mathcal{H}_0(S)) \wedge h^*(z) = 0\}$.*

*Proof.* (Proof of Theorem B.3) The construction of the robustly-reliable learner for $\ell_{IA}$ is similar to the robustly-reliable learner for $\ell_{ST}$. The key difference is that the reliability radius now depends on the predicted label. Given sample $S$, consider the learner $\mathcal{L}$ which outputs $h_S^{\mathcal{L}} = \text{argmin}_{h \in \mathcal{H}} \text{err}_S(h)$.

1. If $h_S^{\mathcal{L}}(z) = 1$, $r_S^{\mathcal{L}}(z)$ is given by the largest $\eta > 0$ for which $\mathbf{B}^o(z, \eta) \subseteq \text{Agree}(\mathcal{H}_0(S))$ and $h(x) = h(z)$, $\forall x \in \mathbf{B}^o(z, \eta), h \in \mathcal{H}_0(S)$ and $\eta = 0$ when $z \in \text{Agree}(\mathcal{H}_0(S))$, and $-1$ otherwise. Note that the supremum exists here since a union of open sets is also open.

2. If $h_S^{\mathcal{L}}(z) = 0$, $r_S^{\mathcal{L}}(z) = \infty$ when $z \in \text{Agree}(\mathcal{H}_0(S))$, and $-1$ otherwise.

We first show that $\mathcal{L}$ is robustly-reliable w.r.t. $\mathcal{M}$ for loss $\ell_{IA}$. By realizability, $\text{err}_S(h_S^{\mathcal{L}}) \leq \text{err}_S(h^*) = 0$, or $h_S^{\mathcal{L}} \in \mathcal{H}_0(S)$. For $z \in \mathcal{X}$, if $h_S^{\mathcal{L}}(z) = 1$ and $r_S^{\mathcal{L}}(z) = \eta \geq 0$, then $\mathbf{B}^o(z, \eta) \subseteq \text{Agree}(\mathcal{H}_0(S))$, in particular $z \in \text{Agree}(\mathcal{H}_0(S))$. Moreover, by definition, for any $x \in \mathbf{B}^o(z, \eta)$, we have $h_S^{\mathcal{L}}(z) = h_S^{\mathcal{L}}(x)$ by construction. Putting together, and using the property that distance functions of a metric are symmetric, we have $h_S^{\mathcal{L}}(z) = h^*(x)$ for any $x$ such that $z \in \mathbf{B}^o(x, \eta)$. Thus, we have $\ell_{ST}^{h^*}(h_S^{\mathcal{L}}, x, z) = 0$ for any $x$ such that $z \in \mathbf{B}^o(x, \eta)$. This also implies that $\ell_{IA}^{h^*}(h_S^{\mathcal{L}}, x, z) = 0$ since $\ell_{ST}$ implies $\ell_{IA}$.

On the other hand, if $h_S^{\mathcal{L}}(z) = 0$ and $r_S^{\mathcal{L}}(z) = \infty$, we have $z \in \text{Agree}(\mathcal{H}_0(S))$. This implies that $h_S^{\mathcal{L}}(z) = h^*(z) = 0$. For any $x$ that $z \in \mathcal{U}_{IA}(x, h^*)$, by the incentive-aware property of the adversary, if $h^*(x) = 1$, we must have $\mathcal{U}_{IA}(x, h^*) = \{x\}$ which implies that $z = x$ and $h^*(z) = h^*(x) = 1$. In our case, $h^*(z) = 0$ also implies that we must also have $h^*(x) = 0$. Therefore, $\ell_{IA}^{h^*}(h_S^{\mathcal{L}}, x, z) = \mathbb{I}[h_S^{\mathcal{L}}(z) \neq h^*(x) \wedge z \in \mathcal{U}_{IA}(x, h^*)] \leq \mathbb{I}[h_S^{\mathcal{L}}(z) \neq h^*(x)] = 0$. Therefore, we can conclude that $\mathcal{L}$ satisfies Definition 3.

Conversely, we will show that for any robustly-reliable learner $\mathcal{L}$ w.r.t. $\ell_{IA}$, for any $\eta > 0$,

$$RR_{IA}^{\mathcal{L}}(S, \eta) \cap \{z \mid h_S^{\mathcal{L}}(z) = 0\} \subseteq \text{Agree}(\mathcal{H}_0(S)) \cap \{z \mid h^*(z) = 0\}.$$

Since $z \in \mathcal{U}_{IA}(z)$, for $z$ to lie in the robustly-reliable region, we need $\ell_{IA}^{h^*}(h_S^{\mathcal{L}}, z, z) = \mathbb{I}[h_S^{\mathcal{L}}(z) \neq h^*(z)] = 0$ that is $z$ must be reliable. By similar arguments from Theorem B.1, we have the result.

Next, we will show that, for any $\eta > 0$,

$$RR_{IA}^{\mathcal{L}}(S, \eta) \cap \{z \mid h_S^{\mathcal{L}}(z) = 1\} \subseteq \{z \mid \mathbf{B}_{\mathcal{M}}^o(z, \eta) \subseteq \text{Agree}(\mathcal{H}_0(S))\}.$$

We will prove this by contradiction. Suppose $z \in \text{Agree}(H_0(S))$, but there exists $x' \in \mathbf{B}_{\mathcal{M}}^o(z, \eta)$ such that $x' \notin \text{Agree}(H_0(S))$. Let there be a robustly-reliable learner $\mathcal{L}$ such that $z \in RR_{IA}^{\mathcal{L}}(S, \eta)$ and $h_S^{\mathcal{L}}(z) = 1$. Because $x' \notin \text{Agree}(H_0(S))$, there exists $h_1 \in H_0(S)$ such that $h_1(x') = 0$. We may have $h^* = h_1$ since $h_1$ is consistent with $S$. However, we have $\mathcal{U}_{IA}(x', h_1) = \mathbf{B}_{\mathcal{M}}^o(x', \eta)$ and

$$\ell_{IA}^{h^*}(h_S^{\mathcal{L}}, x', z) = \mathbb{I}[h(z) \neq h^*(x') \wedge z \in \mathcal{U}_{IA}(x')] = 1$$

which contradicts with $z$ lies in the robustly-reliable region. Furthermore, with a similar argument that we can't have $h^*(x') = 0$, we can show that the agreed label of any $x$ must be 1,

$$RR_{IA}^{\mathcal{L}}(S, \eta) \cap \{z \mid h_S^{\mathcal{L}}(z) = 1\}$$
$$\subseteq \{z \mid \mathbf{B}_{\mathcal{M}}^o(z, \eta) \subseteq \text{Agree}(\mathcal{H}_0(S)) \wedge h(x) = 1, \forall x \in \mathbf{B}_{\mathcal{M}}^o(z, \eta), h \in \mathcal{H}_0(S)\}.$$
$$= A_{ST} \cap \{z \mid h^*(z) = 1\}$$

This concludes that for any robustly-reliable learner $\mathcal{L}$ with respect to $\ell_{IA}$, we have

$$RR_{IA}^{\mathcal{L}}(S, \eta) \subseteq (A_{ST} \cap \{z \mid h^*(z) = 1\}) \cup \{z \mid z \in \text{Agree}(\mathcal{H}_0(S)) \wedge h^*(z) = 0\}.$$

$\square$

# C  General robustly-reliable learner

**Definition 12** (General robustly-reliable learner). *A learner $\mathcal{L}$ is robustly-reliable for sample $S$ w.r.t. a perturbation function $\mathcal{U}$, concept space $\mathcal{H}$ and robust loss function $\ell$ if, for any target concept $h^* \in \mathcal{H}$, given $S$ labeled by $h^*$, the learner outputs functions $h_S^{\mathcal{L}} : \mathcal{X} \to \mathcal{Y}$ and $a_S^{\mathcal{L}} : \mathcal{X} \to \{0, 1\}$ such that for all $z \in \mathcal{X}$ if $a_S^{\mathcal{L}}(z) = 1$ and $z \in \mathcal{U}(x)$ then $\ell^{h^*}(h_S^{\mathcal{L}}, x, z) = 0$. On the other hand, if $a_S^{\mathcal{L}}(z) = 0$, our learner abstains from prediction. The robustly-reliable region of a learner $\mathcal{L}$ is defined as $RR^{\mathcal{L}}(S) = \{x \in \mathcal{X} \mid a_S^{\mathcal{L}}(x) = 1\}$, the region that the learner $\mathcal{L}$ does not abstain.*

We again obtain the pointwise optimal characterization of the robustly-reliable region in terms of the agreement region. For $\ell_{CA}, \ell_{TL}$ the robustly-reliable region would be the same as the region where we can be sure of what the correct label is: i.e. the agreement region of the version space while for $\ell_{ST}$, it is the region of points $z$ for which $\mathcal{U}^{-1}(z)$ lies inside the agreement region of the version space, and all classifiers in the version space agree on $\mathcal{U}^{-1}(z)$.

**Theorem C.1.** *Let $\mathcal{H}$ be any hypothesis class, and $\mathcal{U}$ be the perturbation function.*

(a) *There exists a robustly-reliable learner $\mathcal{L}$ w.r.t. $\mathcal{U}$ and $\ell_{CA}$ such that $RR_{CA}^{\mathcal{L}}(S) \supseteq \mathrm{Agree}(\mathcal{H}_0(S))$. Moreover, for any robustly-reliable learner $\mathcal{L}$, $RR_{CA}^{\mathcal{L}}(S) \subseteq \mathrm{Agree}(\mathcal{H}_0(S))$.*

(b) *The same results hold for $RR_{TL}^{\mathcal{L}}$ as well.*

(c) *There exists a robustly-reliable learner $\mathcal{L}$ w.r.t. $\mathcal{U}$ and $\ell_{ST}$, such that $RR_{ST}^{\mathcal{L}}(S) \supseteq A_{ST}$, and for any $\mathcal{L}$ robustly-reliable w.r.t. $\ell_{ST}$, $RR_{ST}^{\mathcal{L}}(S) \subseteq A_{ST}$, where $A_{ST} = \{z \mid \mathcal{U}^{-1}(z) \subseteq \mathrm{Agree}(\mathcal{H}_0(S)) \wedge h(x) = h(z), \forall x \in \mathcal{U}^{-1}(z), h \in \mathcal{H}_0(S)\}$.*

(d) *There exists a robustly-reliable learner $\mathcal{L}$ w.r.t. $\mathcal{U}$ and $\ell_{IA}$, such that $RR_{IA}^{\mathcal{L}}(S) \supseteq A_{IA}$, and for any $\mathcal{L}$ robustly-reliable w.r.t. $\ell_{IA}$, $RR_{IA}^{\mathcal{L}}(S) \subseteq A_{IA}$, where $A_{IA} = (A_{ST} \cap \{z \mid h^*(z) = 1\}) \cup \{z \mid z \in \mathrm{Agree}(\mathcal{H}_0(S)) \wedge h^*(z) = 0\}$.*

*Proof.* We first establish part (a). Given sample $S$, consider the learner $\mathcal{L}$ which outputs $h_S^{\mathcal{L}} = \mathrm{argmin}_{h \in \mathcal{H}} \mathrm{err}_S(h)$ i.e. an ERM over $S$, and $a_S^{\mathcal{L}}(z) = \mathbb{I}[z \in \mathrm{Agree}(\mathcal{H}_0(S))]$. By realizability, $\mathrm{err}_S(h_S^{\mathcal{L}}) \le \mathrm{err}_S(h^*) = 0$, or $h_S^{\mathcal{L}} \in \mathcal{H}_0(S)$. We first show that $\mathcal{L}$ is robustly-reliable. For $z \in \mathcal{X}$, if $a_S^{\mathcal{L}}(z) = 1$, then $z \in \mathrm{Agree}(\mathcal{H}_0(S))$. We have $h^*(z) = h_S^{\mathcal{L}}(z)$ since the classifiers $h^*, h_S^{\mathcal{L}} \in \mathcal{H}_0(S)$ and $z$ lies in the agreement region of classifiers in $\mathcal{H}_0(S)$. Thus, we have $\ell_{CA}^{h^*}(h_S^{\mathcal{L}}, x, z) = 0$ for any $x$ such that $z \in \mathcal{U}(x)$. $RR_{CA}^{\mathcal{L}}(S) \supseteq \mathrm{Agree}(\mathcal{H}_0(S))$ follows from the choice of $a_S^{\mathcal{L}}(z) = \mathbb{I}[z \in \mathrm{Agree}(\mathcal{H}_0(S))]$.

On the other hand, Let $z \in \mathrm{DIS}(\mathcal{H}_0(S))$. There exist $h_1, h_2 \in \mathcal{H}_0(S)$ such that $h_1(z) \ne h_2(z)$. If possible, let there be a robustly-reliable learner $\mathcal{L}$ such that $z \in RR_{CA}^{\mathcal{L}}(S)$. That is, $a_S^{\mathcal{L}}(z) = 1$. We have $z \in \mathcal{U}(z)$, and therefore $\ell_{CA}^{h^*}(h_S^{\mathcal{L}}, z, z) = 0$. But then we must have $h_S^{\mathcal{L}}(z) = h^*(z)$ by definition of $\ell_{CA}$. We can set $h^* = h_1$ or $h^* = h_2$ since both are consistent with $S$. However, $h_1(z) \ne h_2(z)$, and therefore $h_S^{\mathcal{L}}(z) \ne h^*(z)$ for one of the above choices for $h^*$, contradicting that $\mathcal{L}$ is robustly-reliable.

This completes the proof of (a). Essentially the same argument may be used to establish (b), by substituting $\ell_{CA}$ with $\ell_{TL}$. We will now turn our attention to part (c).

Given sample $S$, consider the learner $\mathcal{L}$ which outputs $h_S^{\mathcal{L}} = \mathrm{argmin}_{h \in \mathcal{H}} \mathrm{err}_S(h)$, that is an ERM over $S$, and $a_S^{\mathcal{L}}(z) = \mathbb{I}[\mathcal{U}^{-1}(z) \in \mathrm{Agree}(\mathcal{H}_0(S)) \wedge h_S^{\mathcal{L}}(x_1) = h_S^{\mathcal{L}}(x_2) \forall x_1, x_2 \in \mathcal{U}^{-1}(z)]$. By realizability, $\mathrm{err}_S(h_S^{\mathcal{L}}) \le \mathrm{err}_S(h^*) = 0$, or $h_S^{\mathcal{L}} \in \mathcal{H}_0(S)$. We first show that $\mathcal{L}$ is robustly-reliable w.r.t. $\ell_{ST}$. For $z \in \mathcal{X}$, if $a_S^{\mathcal{L}}(z) = 1$, then $\mathcal{U}^{-1}(z) \subseteq \mathrm{Agree}(\mathcal{H}_0(S))$, in particular $z \in \mathrm{Agree}(\mathcal{H}_0(S))$. Moreover, by definition, for any $x$ that $z \in U(x)$, we have $h_S^{\mathcal{L}}(z) = h_S^{\mathcal{L}}(x)$ by construction of $a_S^{\mathcal{L}}(z)$. Putting together, we have $h_S^{\mathcal{L}}(z) = h^*(x)$ for any $x$ such that $z \in \mathcal{U}(x)$. Thus, we have $\ell_{ST}^{h^*}(h_S^{\mathcal{L}}, x, z) = 0$ for any $x$ such that $z \in \mathcal{U}(x)$.

On the other hand, for any robustly-reliable learner $\mathcal{L}$, we will show that

$$RR_{ST}^{\mathcal{L}}(S) \subseteq \mathrm{Agree}(\mathcal{H}_0(S)).$$

Let $z \in \mathrm{DIS}(\mathcal{H}_0(S))$. There exists $h_1, h_2 \in \mathcal{H}_0(S)$ such that $h_1(z) \ne h_2(z)$. If possible, let there be a robustly-reliable learner $\mathcal{L}$ such that $z \in RR_{ST}^{\mathcal{L}}(S)$. That is, $a_S^{\mathcal{L}}(z) = 1$. We have $z \in \mathcal{U}^{-1}(z)$,

and therefore $\ell_{\mathrm{ST}}^{h^*}(h_S^{\mathcal{L}}, z, z) = \mathbb{I}[h_S^{\mathcal{L}}(z) \neq h^*(z)] = 0$ which implies that $h_S^{\mathcal{L}}(z) = h^*(z)$. But we can set $h^* = h_1$ or $h^* = h_2$ since both are consistent with $S$. However, $h_1(z) \neq h_2(z)$, and therefore $h_S^{\mathcal{L}}(z) \neq h^*(z)$ for one of the above choices for $h^*$, contradicting that $\mathcal{L}$ is robustly-reliable. Next, we will show that

$$\mathrm{RR}_{\mathrm{ST}}^{\mathcal{L}}(S) \subseteq \{z \mid \mathcal{U}^{-1}(z) \subseteq \mathrm{Agree}(\mathcal{H}_0(S))\}.$$

We will prove this by contradiction, $z \in \mathrm{Agree}(H_0(S))$ but there exists $x' \in \mathcal{U}^{-1}(z)$ such that $x' \notin \mathrm{Agree}(H_0(S))$. Let there be a robustly-reliable learner $\mathcal{L}$ such that $z \in \mathrm{RR}_{\mathrm{ST}}^{\mathcal{L}}(S)$. By definition, we have $\ell_{\mathrm{ST}}^{h^*}(h_S^{\mathcal{L}}, x, z) = 0$ for any $x$ that $z \in \mathcal{U}(x)$. This implies that $\ell_{\mathrm{ST}}^{h^*}(h_S^{\mathcal{L}}, x', z) = 0$ that is $h_S^{\mathcal{L}}(z) = h^*(x')$. Because $x' \notin \mathrm{Agree}(H_0(S))$, there exists $h_1, h_2 \in H_0(S)$ such that $h_1(x') \neq h_2(x')$. We can set $h^* = h_1$ or $h^* = h_2$ since both are consistent with $S$. But $h_1(x') \neq h_2(x')$, and therefore $h_S^{\mathcal{L}}(z) \neq h^*(z)$ for one of the above choices for $h^*$, contradicting that $\mathcal{L}$ is robustly-reliable. Next, we will show that

$$\mathrm{RR}_{\mathrm{ST}}^{\mathcal{L}}(S) \subseteq \{z \mid \mathcal{U}^{-1}(z) \subseteq \mathrm{Agree}(\mathcal{H}_0(S)) \wedge h(x) = h(z),\ \forall\, x \in \mathcal{U}^{-1}(z), h \in \mathcal{H}_0(S)\}.$$

Let $z$ be a data point that $\mathcal{U}^{-1}(z) \subseteq \mathrm{Agree}(H_0(S))$ but there exists $x' \in \mathcal{U}^{-1}(z)$ that $h(x') \neq h(z)$ for $h \in H_0(S)$. Let there be a robustly-reliable learner that $z \in \mathrm{RR}_{\mathrm{ST}}^{\mathcal{L}}(S)$. This implies that $\ell_{\mathrm{ST}}(h_S^{\mathcal{L}}, x, z) = 0$ for any $x$ that $z \in \mathcal{U}(x)$. However, $\ell_{\mathrm{ST}}(h_S^{\mathcal{L}}, x', z) = \mathbb{I}[h_S^{\mathcal{L}}(z) \neq h^*(x')] = \mathbb{I}[h_S^{\mathcal{L}}(z) \neq h_S^{\mathcal{L}}(x')] \neq 0$, contradicting that $\mathcal{L}$ is robustly-reliable.

Finally, the proof of part d) is similar to the proof of part c). Given sample $S$, consider the learner $\mathcal{L}$ which outputs $h_S^{\mathcal{L}} = \mathrm{argmin}_{h \in \mathcal{H}} \mathrm{err}_S(h)$, that is an ERM over $S$, and

1. if $h_S^{\mathcal{L}}(z) = 1$, let $a_S^{\mathcal{L}}(z) = \mathbb{I}[\mathcal{U}^{-1}(z) \in \mathrm{Agree}(\mathcal{H}_0(S)) \wedge h_S^{\mathcal{L}}(x_1) = h_S^{\mathcal{L}}(x_2)\ \forall\ x_1, x_2 \in \mathcal{U}^{-1}(z)]$;

2. if $h_S^{\mathcal{L}}(z) = 0$, let $a_S^{\mathcal{L}}(z) = \mathbb{I}[z \in \mathrm{Agree}(\mathcal{H}_0(S))]$.

By realizability, $\mathrm{err}_S(h_S^{\mathcal{L}}) \leq \mathrm{err}_S(h^*) = 0$, or $h_S^{\mathcal{L}} \in \mathcal{H}_0(S)$. We first show that $\mathcal{L}$ is robustly-reliable w.r.t. $\ell_{\mathrm{IA}}$. For $z \in \mathcal{X}$, if $h_S^{\mathcal{L}}(z) = 1$ and $a_S^{\mathcal{L}}(z) = 1$, then $\mathcal{U}^{-1}(z) \subseteq \mathrm{Agree}(\mathcal{H}_0(S))$, in particular $z \in \mathrm{Agree}(\mathcal{H}_0(S))$. Moreover, by definition, for any $x$ that $z \in U(x)$, we have $h_S^{\mathcal{L}}(z) = h_S^{\mathcal{L}}(x)$ by construction of $a_S^{\mathcal{L}}(z)$. Putting together, we have $h_S^{\mathcal{L}}(z) = h^*(x)$ for any $x$ such that $z \in \mathcal{U}(x)$. Thus, we have $\ell_{\mathrm{ST}}^{h^*}(h_S^{\mathcal{L}}, x, z) = 0$ for any $x$ such that $z \in \mathcal{U}(x)$. This also implies that $\ell_{\mathrm{IA}}^{h^*}(h_S^{\mathcal{L}}, x, z) = 0$ for any $x$ such that $z \in \mathcal{U}(x)$ since $\ell_{\mathrm{ST}}$ implies $\ell_{\mathrm{IA}}$.

For $z \in \mathcal{X}$, if $h_S^{\mathcal{L}}(z) = 0$ and $a_S^{\mathcal{L}}(z) = 1$, we have $z \in \mathrm{Agree}(\mathcal{H}_0(S))$ and $h_S^{\mathcal{L}}(z) = h^*(z) = 0$. By the incentive-aware property of the adversary, any $x$ such that $z \in \mathcal{U}(x)$, we can't have $h^*(x) = 1$ since the adversary has no incentive to make any perturbation in this case. Therefore, we have $h^*(x) = 0$ and $\mathcal{U}_{\mathrm{IA}}(h^*, x) = \mathcal{U}(x)$. We have $\ell_{\mathrm{IA}}^{h^*}(h_S^{\mathcal{L}}, x, z) = \mathbb{I}[h(z) \neq h^*(x) \wedge z \in \mathcal{U}_{\mathrm{IA}}(x, h^*)] = 0$. We can conclude that our learner $\mathcal{L}$ is robustly-reliable w.r.t. $\ell_{\mathrm{IA}}$.

Conversely, we will show that for any robustly-reliable learner $\mathcal{L}$ w.r.t. $\ell_{\mathrm{IA}}$, for any $\eta > 0$,

$$\mathrm{RR}_{\mathrm{IA}}^{\mathcal{L}}(S) \cap \{z \mid h_S^{\mathcal{L}}(z) = 0\} \subseteq \mathrm{Agree}(\mathcal{H}_0(S)) \cap \{z \mid h^*(z) = 0\}.$$

Since $z \in \mathcal{U}_{\mathrm{IA}}(z)$, for $z$ to lie in the robustly-reliable region, we need $\ell_{\mathrm{IA}}^{h^*}(h_S^{\mathcal{L}}, z, z) = \mathbb{I}[h_S^{\mathcal{L}}(z) \neq h^*(z)] = 0$ that is $z$ must be reliable. By similar arguments from above, we have the result. Next, we will show that,

$$\mathrm{RR}_{\mathrm{IA}}^{\mathcal{L}}(S) \cap \{z \mid h_S^{\mathcal{L}}(z) = 1\} \subseteq \{z \mid \mathcal{U}^{-1}(z) \subseteq \mathrm{Agree}(\mathcal{H}_0(S))\}.$$

We will prove this by contradiction. Suppose $z \in \mathrm{Agree}(H_0(S))$, but there exists $x' \in \mathcal{U}^{-1}(z)$ such that $x' \notin \mathrm{Agree}(H_0(S))$. Let there be a robustly-reliable learner $\mathcal{L}$ such that $z \in \mathrm{RR}_{\mathrm{IA}}^{\mathcal{L}}(S)$ and $h_S^{\mathcal{L}}(z) = 1$. Because $x' \notin \mathrm{Agree}(H_0(S))$, there exists $h_1 \in H_0(S)$ such that $h_1(x') = 0$. We may have $h^* = h_1$ since $h_1$ is consistent with $S$. However, we have $\mathcal{U}_{\mathrm{IA}}(x', h_1) = \mathcal{U}(x')$ and

$$\ell_{\mathrm{IA}}^{h^*}(h_S^{\mathcal{L}}, x', z) = \mathbb{I}[h(z) \neq h^*(x') \wedge z \in \mathcal{U}_{\mathrm{IA}}(x')] = 1$$

which contradicts with $z$ lies in the robustly-reliable region. Furthermore, with a similar argument that we can't have $h^*(x') = 0$, we can show that the agreed label of any $x$ must be 1,

$$\mathrm{RR}_{\mathrm{IA}}^{\mathcal{L}}(S, \eta) \cap \{z \mid h_S^{\mathcal{L}}(z) = 1\}$$
$$\subseteq \{z \mid \mathcal{U}^{-1}(z) \subseteq \mathrm{Agree}(\mathcal{H}_0(S)) \wedge h(x) = 1,\ \forall\, x \in \mathcal{U}^{-1}(z), h \in \mathcal{H}_0(S)\}.$$
$$= A_{\mathrm{ST}} \cap \{z \mid h^*(z) = 1\}$$

This concludes that for any robustly-reliable learner $\mathcal{L}$ with respect to $\ell_{\mathrm{IA}}$, we have

$$\mathrm{RR}_{\mathrm{IA}}^{\mathcal{L}}(S) \subseteq (A_{\mathrm{ST}} \cap \{z \mid h^*(z) = 1\}) \cup \{z \mid z \in \mathrm{Agree}(\mathcal{H}_0(S)) \wedge h^*(z) = 0\}.$$

$\square$

We can also define a safely-reliable region for general perturbations as follows.

**Definition 13.** *(General safely-reliable region) Let $\mathcal{L}$ be a robustly-reliable learner w.r.t. a perturbation function $\mathcal{U}$ for sample $S$, concept space $\mathcal{H}$ and robust loss function $\ell$. The safely-reliable region of a learner $\mathcal{L}$ is defined as $SR^{\mathcal{L}}(S) = \{x \in \mathcal{X} \mid \mathcal{U}(x) \subseteq RR^{\mathcal{L}}(S)\}$.*

# D   Additional proof details for safely-reliable region

**Lemma D.1.** *Let $\mathcal{D}$ be isotropic log-concave over $\mathbb{R}^d$ and $\mathcal{H} = \{h : x \to \mathrm{sign}(\langle w_h, x \rangle) \mid w_h \in \mathbb{R}^d, \|w_h\|_2 = 1\}$ be the class of linear separators. Let $\mathbf{B}(\cdot, \eta)$ be a $L_2$ ball perturbation with radius $\eta$. For $S \sim \mathcal{D}^m$, for $m = \mathcal{O}(\frac{1}{\varepsilon^2}(\mathrm{VCdim}(\mathcal{H}) + \ln\frac{1}{\delta}))$, with probability at least $1 - \delta$, we have*

$$\Pr(\{x \mid \mathbf{B}(x, \eta) \subseteq \mathrm{Agree}(\mathcal{H}_0(S))\}) \geq 1 - 2\eta - \tilde{\mathcal{O}}(\sqrt{d}\varepsilon).$$

*Proof.* (Proof of Lemma D.1) From uniform convergence (Theorem 4.1 [AB99]), for $S \sim \mathcal{D}^m$, for $m = \mathcal{O}(\frac{1}{\varepsilon^2}(\mathrm{VCdim}(\mathcal{H}) + \ln\frac{1}{\delta}))$, with probability at least $1 - \delta$, we have $\mathrm{Agree}(B_{\mathcal{D}}^{\mathcal{H}}(h^*, \varepsilon)) \subseteq \mathrm{Agree}(\mathcal{H}_0(S))$. From [BL13] (Theorem 14), for linear separators on a log-concave distribution $A = \{x : \|x\|_2 < \alpha\sqrt{d}\} \cap \{x : |\langle w_{h^*}, x \rangle| \geq C_1 \alpha \varepsilon \sqrt{d}\} \subseteq \mathrm{Agree}(B_{\mathcal{D}}^{\mathcal{H}}(h^*, \varepsilon))$ for some constant $C_1$. We claim that for any $x \in A_\eta := \{x : \|x\|_2 < \alpha\sqrt{d} - \eta\} \cap \{x : |\langle w_{h^*}, x \rangle| \geq C_1 \alpha \varepsilon \sqrt{d} + \eta\}$, we have $\mathbf{B}(x, \eta) \subseteq A$. Let $x \in A_\eta$, consider $z \in \mathbf{B}(x, \eta)$. We have $\|z\|_2 \leq \|z - x\|_2 + \|x\|_2 \leq \eta + \alpha\sqrt{d} - \eta = \alpha\sqrt{d}$ and $|\langle w_{h^*}, z \rangle| \geq |\langle w_{h^*}, x \rangle| - |\langle w_{h^*}, z - x \rangle| \geq C_1 \alpha \varepsilon \sqrt{d} + \eta - \|z - x\| \geq C_1 \alpha \varepsilon \sqrt{d}$. Therefore, $z \in A$ for any $z \in \mathbf{B}(x, \eta)$ which implies that for any $x \in A_\eta$, $\mathbf{B}(x, \eta) \subseteq A$ which also implies that $A_\eta \subseteq \{x \mid \mathbf{B}(x, \eta) \subseteq \mathrm{Agree}(\mathcal{H}_0(S))\}$. We can bound the probability mass of $A_\eta$ with the following fact on isotropic log-concave distribution $\mathcal{D}$ over $\mathbb{R}^d$ [LV07]: 1) $\Pr_{x \sim \mathcal{D}}(\|x\| \geq \alpha\sqrt{d}) \leq e^{-\alpha + 1}$, 2) When $d = 1$ $\Pr_{x \sim \mathcal{D}}(x \in [a, b]) \leq |b - a|$ and 3) The projection $\langle w_{h^*}, x \rangle$ follows a 1-dimensional isotropic log-concave distribution. We have $\Pr_{x \sim \mathcal{D}}(A_\eta) \geq 1 - \Pr_{x \sim \mathcal{D}}(\{x : \|x\| \geq \alpha\sqrt{d} - \eta\}) - \Pr_{x \sim \mathcal{D}}(\{x : |\langle w_{h^*}, x \rangle| \leq C_1 \alpha \varepsilon \sqrt{d} + \eta\}) \geq 1 - e^{-\left(\alpha - \frac{\eta}{\sqrt{d}}\right) + 1} - 2C_1 \alpha \varepsilon \sqrt{d} - 2\eta = 1 - 2\eta - \tilde{\mathcal{O}}(\sqrt{d}\varepsilon)$. The final line holds when we set $\alpha = \ln(\frac{1}{\sqrt{d}\varepsilon})$. $\square$

*Proof.* (Proof of Lemma 3.2) First, we will show that for any sample $S \sim \mathcal{D}^m$ with no points lying on the decision boundary of $h^*$, there exists a constant $\delta_1(S)$ such that for any $h$ with a small enough angle to $h^*$, $\theta(w_h, w_{h^*}) \leq \delta_1$, $h$ must have the same prediction as $h^*$ on $S$ that is $h \in \mathcal{H}_0(S)$. Since there is no point lying on the decision boundary, we have $\min_{x \in S} \frac{|\langle w_{h^*}, x \rangle|}{\|x\|} > 0$.

For $\delta_1$, such that $0 < \delta_1 < \min_{x \in S} \frac{|\langle w_{h^*}, x \rangle|}{\|x\|}$, if $\theta(w_h, w_{h^*}) \leq \delta_1$, for any $x \in S$,

$$\begin{aligned}
|\langle w_{h^*}, x \rangle - \langle w_h, x \rangle| &\leq \|w_{h^*} - w_h\| \|x\| \\
&\leq \theta(w_{h^*}, w_h) \|x\| \\
&< |\langle w_{h^*}, x \rangle|.
\end{aligned}$$

The second to last inequality holds due to the fact that the arc length cannot be smaller than corresponding chord length, and the last inequality follows from the assumption $\theta(w_h, w_{h^*}) \leq \delta_1$. This implies that $\langle w_{h^*}, x \rangle \langle w_h, x \rangle > 0$ and $h \in \mathcal{H}_0(S)$. Now, consider any $x$ such that $c \leq \|x\| \leq d$. We will show that there exists a constant $\delta = \delta(S, c, d)$ such that if the margin of $|\langle w_{h^*}, x \rangle|$ is smaller than $\delta$ then $x \notin \mathrm{Agree}(\mathcal{H}_0(S))$. Since $|\langle w_{h^*}, x \rangle| = \|x\| \cos(\theta(w_{h^*}, x)) \geq c \cos(\theta(w_{h^*}, x))$, $\delta > |\langle w_{h^*}, x \rangle|$ implies that $\cos(\theta(w_{h^*}, x)) < \frac{\delta}{c}$ that is the angle between $w_{h^*}$ and $x$ is almost $\frac{\pi}{2}$. Intuitively, we claim that if $\delta$ is small enough then there exists $h \in \mathcal{H}_0(S)$ such that $h(x) \neq h^*(x)$. Without loss of generality, let $\langle w_{h^*}, x \rangle > 0$ ($\theta(w_{h^*}, x) < \frac{\pi}{2}$). We will show that if $\theta(w_{h^*}, x)$ is close enough to $\frac{\pi}{2}$, we can rotate $w_{h^*}$ to $w_h$ with a small enough angle so that $\theta(w_h, w_{h^*}) \leq \delta_1$ but

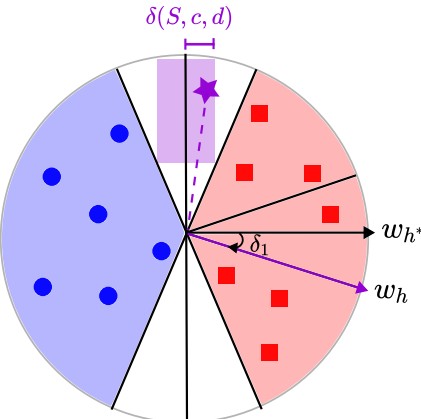

Figure 5: For any set of samples $S$ (blue and red points) with no point lying on the decision boundary of a linear separator $h^*$, for any $0 < c < d$, there exists an area around the decision boundary of $h^*$ (formally defined as $\{x \in \mathbb{R}^d \mid c \leq \|x\| \leq d, |\langle w_{h^*}, x \rangle| \leq \delta(S, c, d)\}$, illustrated by a purple rectangle) such that for any point (purple star) in this area, there exists a hypothesis $h$ that agree with $h^*$ on $S$ but disagree with $h^*$ at that point

$\langle w_h, x \rangle < 0$ ($\theta(w_h, x) > \frac{\pi}{2}$) as illustrated in Figure 5. Formally, we consider $w_h = \frac{w_{h^*} - \lambda x}{\|w_{h^*} - \lambda x\|}$ for some $\lambda > 0$ (to be specified). We will show that there exists $\lambda$ such that 1) $\langle w_h, x \rangle < 0$, and 2) $\theta(w_h, w_{h^*}) \leq \delta_1$. The first condition corresponds to $\lambda > \frac{\langle x, w_{h^*} \rangle}{\|x\|^2}$. The second condition leads to the following inequality

$$\frac{\langle w_{h^*} - \lambda x, w_{h^*} \rangle}{\|w_{h^*} - \lambda x\|} \geq \cos(\delta_1)$$

$$\frac{1 - \lambda \langle x, w_{h^*} \rangle}{\sqrt{1 - 2\lambda \langle x, w_{h^*} \rangle + \|x\|^2 \lambda^2}} \geq \cos(\delta_1)$$

Assume that $\lambda < \frac{1}{\langle x, w_{h^*} \rangle}$, the inequality is equivalent to

$$(1 - \lambda \langle x, w_{h^*} \rangle)^2 \geq \cos^2(\delta_1)(1 - 2\lambda \langle x, w_{h^*} \rangle + \|x\|^2 \lambda^2)$$

$$(\cos^2(\delta_1)\|x\|^2 - \langle x, w_{h^*} \rangle^2)\lambda^2 + 2\lambda \langle x, w_{h^*} \rangle \sin^2(\delta_1) - \sin^2(\delta_1) \leq 0.$$

Solving this inequality leads to

$$\lambda \leq \lambda_{\max} = \frac{-2\sin^2(\delta_1)\langle x, w_{h^*} \rangle + 2\sin(\delta_1)\cos(\delta_1)\sqrt{(\|x\|^2 - \langle x, w_{h^*} \rangle^2)}}{2(\cos^2(\delta_1)\|x\|^2 - \langle x, w_{h^*} \rangle^2)}.$$

Therefore, there exists $\lambda$ that satisfies both of conditions 1), 2) if $\lambda_{\max} > \frac{\langle x, w_{h^*} \rangle}{\|x\|^2}$. Finally, we claim that if $|\langle x, w_{h^*} \rangle| \leq \delta(S, c, d) \leq \frac{c^2 \tan(\delta_1)}{\sqrt{(d + d\tan(\delta_1))^2 + (c^2 \tan(\delta_1))^2}}$ then $\lambda_{\max} > \frac{\langle x, w_{h^*} \rangle}{\|x\|^2}$. For $x$ with $|\langle x, w_{h^*} \rangle| \leq \delta$, we have $\frac{\langle x, w_{h^*} \rangle}{\|x\|^2} \leq \frac{\delta}{\|x\|^2} \leq \frac{\delta}{c^2}$ and also

$$\lambda_{\max} = \frac{-2\sin^2(\delta_1)\langle x, w_{h^*} \rangle + 2\sin(\delta_1)\cos(\delta_1)\sqrt{(\|x\|^2 - \langle x, w_{h^*} \rangle^2)}}{2(\cos^2(\delta_1)\|x\|^2 - \langle x, w_{h^*} \rangle^2)}$$

$$> \frac{-\sin^2(\delta_1)\langle x, w_{h^*} \rangle + \sin(\delta_1)\cos(\delta_1)\sqrt{(\|x\|^2 - \langle x, w_{h^*} \rangle^2)}}{\cos^2(\delta_1)\|x\|^2}$$

$$= \frac{-\sin^2(\delta_1)\frac{\langle x, w_{h^*} \rangle}{\|x\|^2} + \sin(\delta_1)\cos(\delta_1)\sqrt{(1 - (\frac{\langle x, w_{h^*} \rangle}{\|x\|})^2)}}{\cos^2(\delta_1)}$$

$$\geq \frac{-\sin^2(\delta_1)\frac{\delta}{c^2} + \sin(\delta_1)\cos(\delta_1)\sqrt{(1 - (\frac{\delta}{c^2})^2)}}{\cos^2(\delta_1)d}.$$

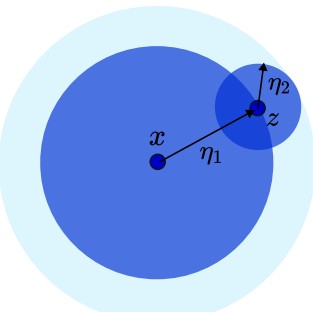

Figure 6: The safely-reliable region contains any point that retains a reliability radius of at least $\eta_2$ even after being attacked by an adversary with strength $\eta_1$.

The last inequality follows from $\frac{\langle x, w_{h^*}\rangle}{\|x\|^2} \leq \frac{\delta}{c^2}$. It is sufficient to show that

$$\frac{-\sin^2(\delta_1)\frac{\delta}{c^2} + \sin(\delta_1)\cos(\delta_1)\sqrt{(1-(\frac{\delta}{c^2})^2)}}{\cos^2(\delta_1)d} \geq \frac{\delta}{c^2}$$

$$\Leftrightarrow \quad -\tan^2(\delta_1)\frac{\delta}{c^2} + \tan(\delta_1)\sqrt{(1-(\frac{\delta}{c^2})^2)} \geq d\frac{\delta}{c^2}$$

$$\Leftrightarrow \quad \tan(\delta_1)\sqrt{(1-(\frac{\delta}{c^2})^2)} \geq (d + \tan^2(\delta_1))\frac{\delta}{c^2}$$

$$\Leftrightarrow \quad \delta(S,c,d) \leq \frac{c^2\tan(\delta_1)}{\sqrt{(d+\tan^2(\delta_1))^2 + c^2\tan^2(\delta_1)}}.$$

$\square$

*Proof.* (Proof of Theorem 3.3) We know that for $\ell_{\mathrm{CA}}, \ell_{\mathrm{TL}}$, the robustly-reliable region is the same as the reliable region. This is also a reason why the probability mass does not depend on $\eta_2$. Consider the optimal robustly-reliable learner $\mathcal{L}$, we have $\mathrm{RR}^{\mathcal{L}}_{\mathrm{CA}}(S,\eta_2) = \mathrm{RR}^{\mathcal{L}}_{\mathrm{TL}}(S,\eta_2) = \mathrm{Agree}(\mathcal{H}_0(S))$ (Theorem B.1). On the other hand, $\mathrm{RR}^{\mathcal{L}}_{\mathrm{ST}}(S,\eta_2) = \{z \mid \mathbf{B}(z,\eta_2) \subseteq \mathrm{Agree}(\mathcal{H}_0(S)) \wedge h(x) = h(z), \forall x \in \mathbf{B}(z,\eta_2), h \in \mathcal{H}_0(S)\}$ (Theorem B.2). **a)** Since $\mathrm{SR}^{\mathcal{L}}_{\mathrm{TL}}(S,\eta_1,\eta_2) = \{x \mid \mathbf{B}(x,\eta_1) \subseteq \mathrm{Agree}(\mathcal{H}_0(S))\}$. Applying Lemma D.1, we have the first result. **b)** Recall that $\mathrm{SR}^{\mathcal{L}}_{\mathrm{CA}}(S,\eta_1,\eta_2) = \{x \in \mathcal{X} \mid \mathbf{B}(x,\eta_1) \cap \{z \mid h^*(z) = h^*(x)\} \subseteq \mathrm{Agree}(\mathcal{H}_0(S))\}$. We will show that for any $x \in \mathrm{SR}^{\mathcal{L}}_{\mathrm{CA}}(S,\eta_1,\eta_2)$, $\mathbf{B}(x,\eta_1) = \mathbf{B}(x,\eta_1) \cap \{z \mid h^*(z) = h^*(x)\}$ by contradiction. Let $x \in \mathrm{SR}^{\mathcal{L}}_{\mathrm{CA}}(S,\eta_1,\eta_2)$ and $\mathbf{B}(x,\eta_1)$ contain two points with a different label, this implies that this ball must contain the decision boundary of $h^*$ ($\mathbf{B}(x,\eta_1) \cap \{x \mid \langle w^*_h, x\rangle = 0\} \neq \emptyset$). The ball must also contain a point that has the same label as $x$ with an arbitrarily small margin w.r.t. $h^*$. For any $a > 0$, there exists $z \in \mathbf{B}(x,\eta_1) \cap \{z \mid h^*(z) = h^*(x)\}$ with $|\langle z, w_{h^*}\rangle| < a$. This is impossible because by Lemma 3.2 the agreement region $\mathrm{Agree}(\mathcal{H}_0(S))$ can not contain point with arbitrarily small margin if $S$ does not contain any point on the decision boundary of $h^*$. This event has a probability 1 as the projection $\langle w_{h^*}, x\rangle$ also follows a log-concave distribution which implies that $\Pr(\langle w_{h^*}, x\rangle = 0) = 0$. Therefore, with probability 1, $\mathbf{B}(x,\eta_1)$ must contain points with the same label and we can conclude that $\mathrm{SR}^{\mathcal{L}}_{\mathrm{CA}}(S,\eta_1,\eta_2) = \mathrm{SR}^{\mathcal{L}}_{\mathrm{TL}}(S,\eta_1,\eta_2)$. **c)** Similarly, by Lemma 3.2, we can show that if $\mathbf{B}(z,\eta_2) \subseteq \mathrm{Agree}(\mathcal{H}_0(S))$, $\mathbf{B}(z,\eta_2)$ then every point in $\mathbf{B}(z,\eta_2)$ must have same label with probability 1. Therefore, $\mathrm{RR}^{\mathcal{L}}_{\mathrm{ST}}(S,\eta_2) = \{z \mid \mathbf{B}(z,\eta_2) \subseteq \mathrm{Agree}(\mathcal{H}_0(S))\}$. We have $\mathrm{SR}^{\mathcal{L}}_{\mathrm{ST}}(S,\eta_1,\eta_2) = \{x \in \mathcal{X} \mid \mathbf{B}_{\mathcal{M}}(x,\eta_1) \subseteq \{z \mid \mathbf{B}(z,\eta_2) \subseteq \mathrm{Agree}(\mathcal{H}_0(S))\}\} = \{x \in \mathcal{X} \mid \mathbf{B}_{\mathcal{M}}(x,\eta_1+\eta_2) \subseteq \mathrm{Agree}(\mathcal{H}_0(S))\}$ by a triangle inequality (see Figure 6). Applying Lemma D.1, we have the result. **d)** With the result above we have $\mathrm{RR}^{\mathcal{L}}_{\mathrm{IA}}(S,\eta_2) = (\{z \mid \mathbf{B}(z,\eta_2) \subseteq \mathrm{Agree}(\mathcal{H}_0(S))\} \cap \{z \mid h^*(z) = 1\}) \cup (\mathrm{Agree}(\mathcal{H}_0(S)) \cap \{z \mid h^*(z) = 0\})$. Recall that $\mathrm{SR}^{\mathcal{L}}_{\mathrm{IA}}(S,\eta_1,\eta_2) = \{x \in \mathcal{X} \mid h^*(x) = 0 \wedge \mathbf{B}_{\mathcal{M}}(x,\eta_1) \subseteq \mathrm{RR}^{\mathcal{L}}_{\mathrm{IA}}(S,\eta_2)\} \cup \{x \in \mathcal{X} \mid h^*(x) = 1 \wedge x \in \mathrm{RR}^{\mathcal{L}}_{\mathrm{CA}}(S,\eta_2)\}$. Therefore, we have $\mathrm{SR}^{\mathcal{L}}_{\mathrm{IA}}(S,\eta_1,\eta_2) = (\{z \mid \mathbf{B}(z,\eta_2) \subseteq \mathrm{Agree}(\mathcal{H}_0(S))\} \cap \{z \mid h^*(z) = 1\}) \cup (\{z \mid \mathbf{B}(z,\eta_1) \subseteq \mathrm{Agree}(\mathcal{H}_0(S))\} \cap \{z \mid h^*(z) = 0\})$ We can conclude the result by applying Lemma D.1 and symmetry. $\square$

# E  Safely-reliable region for classifiers with smooth boundaries

We also bound the probability mass of the safely-reliable region for more general concept spaces beyond linear separators. Specifically, we consider classifiers with smooth boundaries in the sense of [vdVW96].

**Definition 14** ($\alpha$-norm). *Let $f : C \to \mathbb{R}$ be a function on $C \subset \mathbb{R}^d$, and let $\alpha \in \mathbb{R}^+$. For $k = (k_1, \ldots, k_d) \in \mathbf{Z}^d_{\geq 0}$, let $||k||_1 = \sum_{i=1}^d k_i$ and let $D^k = \frac{\partial^k}{\partial^{k_1} x_1 \ldots \partial^{k_d} x_d}$. We define $\alpha$-norm of $f$ as*

$$||f||_\alpha := \max_{||k||_1 < \lceil \alpha \rceil} \sup_{x \in C} |D^k f(x)| + \max_{||k||_1 = \lceil \alpha \rceil - 1} \sup_{x \neq x' \in C} \frac{|D^k f(x) - D^k f(x')|}{|x - x'|^{\alpha - \lceil \alpha \rceil + 1}}.$$

We define $\alpha$th order smooth functions to be those which have a bounded $\alpha$-norm. More precisely, we define the class of $\alpha$th order smooth functions $F_\alpha^C := \{f \mid ||f||_\alpha \leq C\}$. For example, 1st order smoothness corresponds to Lipschitz continuity. We now define concept classes with smooth classification boundaries.

**Definition 15** (Concepts with Smooth Classification Boundaries, [Wan11]). *A set of concepts $\mathcal{H}_\alpha^C$ defined on $\mathcal{X} = [0,1]^{d+1}$ is said to have $\alpha$th order smooth classification boundaries, if for every $h \in \mathcal{H}_\alpha^C$ the classification boundary is the graph of function $x_{d+1} = f(x_1, \ldots, x_d)$, where $f \in F_\alpha^C$ and $(x_1, \ldots, x_{d+1}) \in \mathcal{X}$ i.e. the predicted label is given by $\operatorname{sign}(x_{d+1} - f(x_1, \ldots, x_d))$.*

If we further assume that the probability density may be upper and lower bounded by some absolute positive constants (i.e. "nearly" uniform density), we can bound the safely-reliable region of our learner even in this setting. We start with analogues of Lemmas D.1 and 3.2 for concepts with smooth classification boundaries.

We first bound the probability mass of points $x$ for which $\mathbf{B}(x, \eta)$ is contained in the agreement region of sample-consistent classifiers. We use the Lipschitzness of smooth functions to show that such point $x$ must lie outside of a 'ribbon' around the boundary of target concept $h^*$, and adapt and extend the arguments of [Wan11] to bound the probability mass of this ribbon.

**Lemma E.1.** *Let the instance space be $\mathcal{X} = [0,1]^{d+1}$ and $\mathcal{D}$ be a distribution over $\mathcal{X}$ with a "nearly" uniform density where there exist positive constants $0 < a < b$ such that $a \leq p(x) \leq b$ for all $x \in [0,1]^{d+1}$ when $p(x)$ is the probability density of $\mathcal{D}$. Let $\mathcal{H}_\alpha^C$ be the hypothesis space of concepts with smooth classification boundaries with $d < \alpha < \infty$, and $\mathbf{B}(\cdot, \eta)$ be a $L_2$ ball perturbation with radius $\eta$. For $S \sim \mathcal{D}^m$, for $m = \mathcal{O}(\frac{1}{\varepsilon^2}(\operatorname{VCdim}(\mathcal{H}) + \ln \frac{1}{\delta}))$, with probability at least $1 - \delta$, we have*

$$\Pr(\{x \mid \mathbf{B}(x, \eta) \subseteq \operatorname{Agree}(\mathcal{H}_0(S))\}) \geq 1 - 2b(C+1)\eta - \mathcal{O}\left(ba^{-\frac{\alpha}{d+\alpha}} \varepsilon^{\frac{\alpha}{d+\alpha}}\right).$$

*Proof.* By uniform convergence (Theorem 4.1, [AB99]), for $S \sim \mathcal{D}^m$, for $m = \mathcal{O}(\frac{1}{\varepsilon^2}(\operatorname{VCdim}(\mathcal{H}) + \ln \frac{1}{\delta}))$, with probability at least $1 - \delta$, we have $\operatorname{Agree}(B_\mathcal{D}^\mathcal{H}(h^*, \varepsilon)) \subseteq \operatorname{Agree}(\mathcal{H}_0(S))$. Therefore, it suffices to lower bound $\pi := \Pr\{x \mid \mathbf{B}(x, \eta) \subseteq \operatorname{Agree}(B_\mathcal{D}^\mathcal{H}(h^*, \varepsilon))\}$. Let $h^* \in \mathcal{H}_\alpha^C$ be the target concept and denote $\boldsymbol{x} = (x_1, \ldots, x_d) \in [0,1]^d$. Recall that the predicted label of $(\boldsymbol{x}, x_{d+1})$ from $h, h^*$ is given by $\operatorname{sign}(x_{d+1} - f_h(\boldsymbol{x}))$ and $\operatorname{sign}(x_{d+1} - f_{h^*}(\boldsymbol{x}))$ respectively. Therefore, $h, h^*$ would disagree on $(\boldsymbol{x}, x_{d+1})$ when $x_{d+1}$ lies between $f_h(\boldsymbol{x})$ and $f_{h^*}(\boldsymbol{x})$. Denote $\Phi_h(\boldsymbol{x}) = |\int_{f_{h^*}(\boldsymbol{x})}^{f_h(\boldsymbol{x})} p(\boldsymbol{x}, x_{d+1}) dx_{d+1}|$ be the probability mass of points that $h$ disagree with $h^*$ over $(\boldsymbol{x}, x_{d+1})$ for a fixed $\boldsymbol{x} \in [0,1]^d$. With this notation, the probability mass of points $(\boldsymbol{x}, x_{d+1})$ that $h$ and $h^*$ disagree with is given by $\int_{[0,1]^d} |\Phi_h(\boldsymbol{x})| d\boldsymbol{x}$. Furthermore, from $a \leq p(\boldsymbol{x}, x_{d+1}) \leq b$, we know that $a|f_h(\boldsymbol{x}) - f_{h^*}(\boldsymbol{x})| \leq |\Phi_h(\boldsymbol{x})| \leq b|f_h(\boldsymbol{x}) - f_{h^*}(\boldsymbol{x})|$.

Consider $h \in B_\mathcal{D}(h^*, \varepsilon)$, we have $\int_{[0,1]^d} |\Phi_h(\boldsymbol{x})| d\boldsymbol{x} \leq \varepsilon$. This implies that $\int_{[0,1]^d} |f_h(\boldsymbol{x}) - f_{h^*}(\boldsymbol{x})| d\boldsymbol{x} \leq \int_{[0,1]^d} |\frac{\Phi(\boldsymbol{x})}{a}| d\boldsymbol{x} \leq \frac{\varepsilon}{a}$. Since the classification boundaries are assumed to be $\alpha$th order smooth with $\alpha > d$, Lemma 11 of [Wan11] implies that $||f_h - f_{h^*}||_\infty = O\left((\frac{\varepsilon}{a})^{\frac{\alpha}{d+\alpha}}\right)$ where $||g||_\infty := \sup_{\boldsymbol{x} \in [0,1]^d} |g(\boldsymbol{x})|$. Consider

$$1 - \pi = \Pr_{x \sim \mathcal{D}_\mathcal{X}} (\exists z \in \mathbf{B}(x, \eta), \exists h \in B(h^*, \varepsilon), h(z) \neq h^*(z)).$$

Recall that for $z = (\mathbf{z}, z_{d+1})$, $h(z) \neq h^*(z)$ when $z_{d+1}$ lies between $h(\mathbf{z}), h^*(\mathbf{z})$ which implies $f_{h^*}(\mathbf{z}) - |f_{h^*}(\mathbf{z}) - f_h(\mathbf{z})| < z_{d+1} < f_{h^*}(\mathbf{z}) + |f_{h^*}(\mathbf{z}) - f_h(\mathbf{z})|$. Since $z \in \mathbf{B}(x, \eta)$ and the boundary

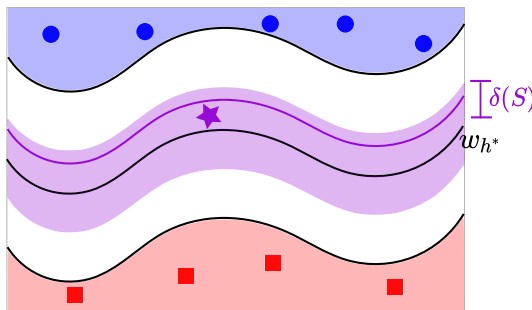

Figure 7: For any set of samples $S$ (blue and red points) with no point lying on the decision boundary of a concept with smooth boundary $h^*$, there exists a band around the decision boundary of $h^*$ (formally defined as $\{x \in \mathbb{R}^d \mid |f_{h^*}(\boldsymbol{x}) - x_{d+1}| < \delta\}$ (purple)) such that for any point (purple star) in this area, there exists a hypothesis $h$ (by translation) that agree with $h^*$ on $S$ but disagree with $h^*$ at that point.

functions $f_h$ are $C$-Lipschitz (by Definition 14) we have $|f_h(\mathbf{z}) - f_h(\boldsymbol{x})| \leq C\|\mathbf{z} - \boldsymbol{x}\| \leq C\eta$ and $|z_{d+1} - x_{d+1}| \leq \eta$. This implies that $f_{h^*}(\boldsymbol{x}) - |f_{h^*}(\mathbf{z}) - f_h(\mathbf{z})| - (C+1)\eta < x_{d+1} < f_{h^*}(\boldsymbol{x}) + |f_{h^*}(\mathbf{z}) - f_h(\mathbf{z})| + (C+1)\eta$. We are interested in the set $\{x \mid \exists z \in \mathbf{B}(x,\eta), \exists h \in B(h^*, \varepsilon), h(z) \neq h^*(z)\}$, this leads to the inequality

$$f_{h^*}(\boldsymbol{x}) - D - (C+1)\eta < x_{d+1} < f_{h^*}(\boldsymbol{x}) + D + (C+1)\eta$$

when $D = \sup_{\substack{z \in \mathbf{B}(x,\eta) \\ h \in B(h^*,\varepsilon)}} |f_{h^*}(\mathbf{z}) - f_h(\mathbf{z})| \leq \sup_{h \in B(h^*,\varepsilon)} \|f_{h^*} - f_h\|_\infty = O\left(\left(\frac{\varepsilon}{a}\right)^{\frac{\alpha}{d+\alpha}}\right)$. Therefore,

$$
\begin{aligned}
1 - \pi &\leq \int_{[0,1]^d} \int_{f_{h^*}(\boldsymbol{x}) - D - (C+1)\eta}^{f_{h^*}(\boldsymbol{x}) + D + (C+1)\eta} p(\boldsymbol{x}, x_{d+1}) dx_{d+1} d\boldsymbol{x} \\
&\leq 2b((C+1)\eta + D) \\
&= 2b(C+1)\eta + O\left(ba^{-\frac{\alpha}{d+\alpha}} \varepsilon^{\frac{\alpha}{d+\alpha}}\right).
\end{aligned}
$$

$\square$

The above bound immediately implies a bound on the probability mass of the safely-reliable region for $\ell_{\text{TL}}$, in combination with our previous results. The following lemma allows us to easily handle the extension of our results to losses $\ell_{\text{CA}}$ and $\ell_{\text{ST}}$ as well, where the safely-reliable region involves additional constraints.

**Lemma E.2.** *Let the instance space be $\mathcal{X} = [0,1]^{d+1}$ and $\mathcal{D}$ be a distribution over $\mathcal{X}$. Let $\mathcal{H}_\alpha^C$ be the hypothesis space of concepts with smooth classification boundaries with $d < \alpha < \infty$. For $h^* \in \mathcal{H}_\alpha^C$, for a set of samples $S \sim \mathcal{D}^m$ such that there is no data point in $S$ that lies on the decision boundary, there exists $\delta(S) > 0$ such that for any $x = (\boldsymbol{x}, x_{d+1})$ with $|f_{h^*}(\boldsymbol{x}) - x_{d+1}| < \delta$, we have $x \notin \text{Agree}(\mathcal{H}_0(S))$.*

*Proof.* Note that translation preserves smoothness, i.e. any concept $h_t$ with $f_{h_t}(\boldsymbol{x}) = f_{h^*}(\boldsymbol{x}) + t$ would also lie in $\mathcal{H}_\alpha^C$. If $|t| < \delta(S) = \min_{x \in S} |f_{h^*}(\boldsymbol{x}) - x_{d+1}|$, $f_{h_t}(\boldsymbol{x}) - x_{d+1}$ must have the same sign as $(f_{h^*}(\boldsymbol{x}) - x_{d+1})$ for any $x \in S$, that is $h_t$ would agree with $h^*$ on every point in $S$. Furthermore, for any $x$ with $|f_{h^*}(\boldsymbol{x}) - x_{d+1}| < \delta$, we can always translate $h^*$ to $h_t$ that $h_t(x) \neq h^*(x)$ (see Figure 7). Therefore, $x \notin \text{Agree}(\mathcal{H}_0(S))$. $\square$

Equipped with the above lemmas, we establish bounds on the probability mass of the safely reliable region for concept classes with smooth classification boundaries.

**Theorem E.3.** *Let the instance space be $\mathcal{X} = [0,1]^{d+1}$ and $\mathcal{D}$ be a distribution over $\mathcal{X}$ with a "nearly" uniform density where there exist positive constants $0 < a < b$ such that $a \leq p(x) \leq b$ for all $x \in [0,1]^{d+1}$ when $p(x)$ is the probability density of $\mathcal{D}$. Let $\mathcal{H}_\alpha^C$ be the hypothesis space of concepts with smooth classification boundaries with $d < \alpha < \infty$, and $\mathbf{B}(\cdot, \eta)$ be a $L_2$ ball perturbation with radius $\eta$. For $S \sim \mathcal{D}^m$, for $m = \mathcal{O}(\frac{1}{\varepsilon^2}(\text{VCdim}(\mathcal{H}) + \ln \frac{1}{\delta}))$ with no point lying on the decision boundary of $h^*$, for an optimal robustly-reliable learner $\mathcal{L}$, we have*

(a) $\Pr(SR_{TL}^{\mathcal{L}}(S, \eta_1, \eta_2)) \geq 1 - 2b(C+1)\eta_1 - \mathcal{O}\left(ba^{-\frac{\alpha}{d+\alpha}}\varepsilon^{\frac{\alpha}{d+\alpha}}\right)$ *with probability* $1 - \delta$,

(b) $SR_{CA}^{\mathcal{L}}(S, \eta_1, \eta_2) = SR_{TL}^{\mathcal{L}}(S, \eta_1, \eta_2)$,

(c) $\Pr(SR_{ST}^{\mathcal{L}}(S, \eta_1, \eta_2)) \geq 1 - 2b(C+1)(\eta_1 + \eta_2) - \mathcal{O}\left(ba^{-\frac{\alpha}{d+\alpha}}\varepsilon^{\frac{\alpha}{d+\alpha}}\right)$ *with probability* $1 - \delta$,

(d) $\Pr(SR_{IA}^{\mathcal{L}}(S, \eta_1, \eta_2)) \geq 1 - b(C+1)(\eta_1 + \eta_2) - \mathcal{O}\left(ba^{-\frac{\alpha}{d+\alpha}}\varepsilon^{\frac{\alpha}{d+\alpha}}\right)$ *with probability* $1 - \delta$.

*Proof.* From Lemma E.2, the agreement region does not contain points that are arbitrarily close to the boundary of $h^*$. Therefore, we can remove the additional conditions on labels for $SR_{CA}^{\mathcal{L}}(S, \eta_1, \eta_2)$ and $SR_{ST}^{\mathcal{L}}(S, \eta_1, \eta_2)$, that is $SR_{CA}^{\mathcal{L}}(S, \eta_1, \eta_2) = SR_{TL}^{\mathcal{L}}(S, \eta_1, \eta_2) = \{x \mid \mathbf{B}(x, \eta_1) \subseteq \text{Agree}(\mathcal{H}_0(S))\}$ and $SR_{ST}^{\mathcal{L}}(S, \eta_1, \eta_2) = \{x \in \mathcal{X} \mid \mathbf{B}_{\mathcal{M}}(x, \eta_1) \subseteq \{z \mid \mathbf{B}(z, \eta_2) \subseteq \text{Agree}(\mathcal{H}_0(S))\}\} = \{x \in \mathcal{X} \mid \mathbf{B}_{\mathcal{M}}(x, \eta_1 + \eta_2) \subseteq \text{Agree}(\mathcal{H}_0(S))\}$. Similarly, we have $SR_{IA}^{\mathcal{L}}(S, \eta_1, \eta_2) = (\{z \mid \mathbf{B}(z, \eta_2) \subseteq \text{Agree}(\mathcal{H}_0(S))\} \cap \{z \mid h^*(z) = 1\}) \cup (\{z \mid \mathbf{B}(z, \eta_1) \subseteq \text{Agree}(\mathcal{H}_0(S))\} \cap \{z \mid h^*(z) = 0\})$. The result follows from Lemma E.1.

$\square$

# F   More on computational efficiency

It is possible to extend the optimization objective to a wide range of hypothesis classes under the following assumption.

**Assumption 1.** *For a hypothesis class $\mathcal{H}$, we assume that for any $h^* \in \mathcal{H}$, a set of data points $S$ labeled by $h^*$ then for any points $x, y$ that $h^*(x) \neq h^*(y)$, the line that connects between $x, y$ must pass through a disagreement region of $\mathcal{H}_0(S)$,*

$$\{\lambda x + (1 - \lambda)y \mid \lambda \in [0, 1]\} \cap \text{DIS}(\mathcal{H}_0(S)) \neq \emptyset.$$

For example, a class of linear separators and a class of classifiers with smooth boundaries satisfies this assumption.

**Lemma F.1.** *Let $\mathcal{H}$ be a hypothesis class and $\mathcal{D}$ be a distribution over $\mathbb{R}^d$. If $\mathcal{H}$ satisfies Assumption 1 then for a set of samples $S \sim \mathcal{D}^m$, the reliability radius of a test point $z$ is given by*

$$\min_{h, h', z'} \|z - z'\|^2$$
$$s.t. \quad h \in \mathcal{H}_0(S),$$
$$h' \in \mathcal{H}_0(S),$$
$$h(z') \neq h'(z').$$

*Proof.* Let $r$ be the largest reliability radius of a test point $z$ that is if we perturb $z$ by a radius at most $r$ then the perturbed point is still in the agreement region. Consider for any perturbation $z'$ that there exists $h, h'$ that $h'(z')$ has a different label from $h'(z)$. If $\mathcal{H}$ satisfies Assumption 1, $h'(z) \neq h'(z')$ implies that the line between $z, z'$ must pass through the disagreement region. Therefore, $r \leq \|z - z'\|$. On the other hand, let $r_0^2$ be the solution of the optimization given above. This implies that for any point $z'$ that $\|z - z'\| < r_0$, for any $h, h' \in \mathcal{H}_0(S)$, we must have that $h(z) = h(z')$ that is $z' \in \text{Agree}(\mathcal{H}_0(S))$. Therefore, we have $r_0 \leq r$ and we can conclude that $r = r_0$. $\square$

**Remark.** This could be solved efficiently in practice. For example, in the case of linear separators, the problem takes the form of solving a quadratic program for each test point. We observe that our proof of Theorem 3.3 above suggests an alternate margin-based approach which might be even more practical to implement, while retaining high probability reliability guarantees under the distributional assumption. If $\hat{h}$ denotes an ERM classifier on sample $S$, one could check the membership $z \in \hat{A}_\eta := \{x : \|x\|_2 < \alpha\sqrt{d} - \eta\} \cap \{x : |\langle w_{\hat{h}}, x\rangle| \geq C_1\alpha\varepsilon\sqrt{d} + \eta\}$ for any $\eta \geq 0$ by just computing the norm and margin w.r.t. $\hat{h}$. A simple halving search (e.g. starting with $\eta = \frac{1}{2}$) could be used to estimate the reliability radius.

Further, we can relax this constrained objective into a regularized objective that can be solved using empirical risk minimization. In the following Lemma, we show that this provides a lower bound on the reliability radius.

**Lemma F.2.** *(Relaxation of the optimization objective for reliability radius) Let $\mathcal{H}$ be a hypothesis class and $\mathcal{D}$ be a distribution over $\mathbb{R}^d$. If $\mathcal{H}$ satisfies Assumption 1 then for a set of samples $S \sim \mathcal{D}^m$, let $h_1, h_2, z^*$ be the optimal solution of the objective*

$$h_1, h_2, z^* = \underset{h, h', z'}{\operatorname{argmin}} \|z - z'\|^2 + \lambda(\hat{R}(h, S \cup \{(z', 0)\}) + \hat{R}(h', S \cup \{(z', 1)\}))$$

*when $\hat{R}(h, A)$ is an empirical risk of $h$ on the sample $A$ then $\|z - z^*\|^2 \le r$ when $r$ the reliability radius of $z$.*

*Proof.* Let $h, h', z'$ be the optimal solution of the optimization objective in Lemma F.1 so that the reliability radius $r$ is given by $\|z - z'\|$. Without loss of generality, let $h(z') = 0$ and $h'(z') = 1$. By definition, we have $\hat{R}(h, S \cup \{(z', 0)\}) = 0$ and $\hat{R}(h', S \cup \{(z', 1)\})) = 0$. Let $h_1, h_2, z^*$ be an optimal solution of the objective in Lemma F.2 then we have

$$\|z - z^*\|^2 + \lambda(\hat{R}(h_1, S \cup \{(z^*, 0)\}) + \hat{R}(h_2, S \cup \{(z^*, 1)\}))$$
$$\le \|z - z'\|^2 + \lambda(\hat{R}(h, S \cup \{(z', 0)\}) + \hat{R}(h', S \cup \{(z', 1)\}))$$
$$= \|z - z'\|^2$$

Since the empirical risk is non-negative, we can conclude that $\|z - z^*\|^2 \le \|z - z'\|^2 = r^2$. $\qquad \square$

Similar observations also apply to the computation of the safely-reliable region. It may be useful to compute the safely-reliable region when we want to estimate the robust reliability performance of an algorithm on test data. In particular, this may be helpful in determining how often and where our learner gives a bad reliability radius, which can inform its safe deployment in practice.

## G   Bounds on the $\mathcal{P} \to \mathcal{Q}$ disagreement coefficient

We will now consider some commonly studied concept spaces, and bound the $\mathcal{P} \to \mathcal{Q}$ disagreement coefficient for broad classes of distribution shifts.

**Linear separators and nearly log-concave or $s$-concave distributions.**   We give a bound on $\Theta_{\mathcal{P} \to \mathcal{Q}}$ for $\mathcal{P}$ and $\mathcal{Q}$ isotropic nearly log-concave distributions [AK91] over $\mathbb{R}^d$, a broad class that includes isotropic log-concave distributions.

**Definition 16** ($\beta$-log-concavity). *A density function $f : \mathbb{R}^d \to \mathbb{R}_{\ge 0}$ is $\beta$-log-concave if for any $\lambda \in [0, 1]$ and any $x_1, x_2 \in \mathbb{R}^d$, we have $f(\lambda x_1 + (1 - \lambda)x_2) \ge e^{-\bar{\beta}} f(x_1)^\lambda f(x_2)^{1-\lambda}$.*

For example, 0-log-concave densities are also log-concave. Also, since the condition in the definition holds for $\lambda = 0$, we have $\beta \ge 0$. Note that a smaller value of $\beta$ makes the distribution closer to logconcave. We have the following bound on $\Theta_{\mathcal{P} \to \mathcal{Q}}$ for distribution shift involving nearly log-concave densities. At a high level, we bound the angle between the normal vectors of the linear separators with small disagreement with $h^*$ under $\mathcal{P}$, and bound the probability mass of their disagreement region under $\mathcal{Q}$, by refining and generalizing the arguments from [BL13]. In particular, we quantify how much near-logconcavity is sufficient for the angle bound to hold, which further implies that any point in the disagreement region is either far away from the mean or close to the margin.

**Theorem G.1.** *Let the concept space $\mathcal{H}$ be the class of linear separators in $\mathbb{R}^d$. Let $\mathcal{P}$ be isotropic $\beta_1$-log-concave and $\mathcal{Q}$ be isotropic $\beta_2$-log-concave, over $\mathbb{R}^d$. Then for $0 \le \beta_1, \beta_2 \le \frac{1}{56\lceil \log_2(d+1)\rceil}$, we have $\Theta_{\mathcal{P} \to \mathcal{Q}}(\varepsilon) = O(d^{1/2 + \frac{\beta_2}{2\ln 2}} \log(d/\varepsilon))$.*

*Proof.* Our proof builds on and generalizes the arguments used in the proof of Theorem 14 in [BL13]. Let $h \in B_{\mathcal{P}}(h^*, r)$, i.e. $d(h, h^*) \le r$. We can apply the whitening transform from Theorem 16 of [BL13] provided $(1/20 + c_1)\sqrt{1/C_1 - c_1^2} \le 1/9$, where $C_1 = e^{\beta_1 \lceil \log_2(d+1)\rceil}$ and

$c_1 = e(C_1 - 1)\sqrt{2C_1}$. It may be verified that this condition holds for $0 \leq \beta_1 \leq \frac{1}{56\lceil \log_2(d+1)\rceil}$. Now, by Theorem 11 of [BL13] we can bound the angle between their normal vectors as $\theta(w_h, w_{h^*}) \leq cr$ where $c$ is an absolute constant. Now if $x \in \mathcal{X}$ has a large margin $|w_{h^*} \cdot x| \geq cr\alpha$ and small norm $||x|| \leq \alpha$, for some $\alpha > 0$, we have

$$|w_h \cdot x - w_{h^*} \cdot x| \leq ||w_h - w_{h^*}|| \cdot ||x|| < cr\alpha.$$

Now the large margin condition $|w_{h^*} \cdot x| \geq cr\alpha$ implies $\langle w_h, x\rangle \langle w_{h^*}, x\rangle > 0$, or $h(x) = h^*(x)$. Since $h \in B_{\mathcal{P}}(h^*, r)$ was arbitrary, we have $x \notin \mathrm{DIS}(B_{\mathcal{P}}(h^*, r)))$. Therefore, the set $\{x \mid ||x|| > \alpha\} \cup \{x \mid |w_{h^*} \cdot x| \leq cr\alpha\}$ contains the disagreement region $\mathrm{DIS}(B_{\mathcal{P}}(h^*, r)))$.

By Theorem 11 of [BL13], since $\mathcal{Q}$ is an isotropic $\beta_2$-log-concave distribution, we have $\mathrm{Pr}_Q[||x|| > R\sqrt{Cd}] < Ce^{-R+1}$, for $C = e^{\beta_2 \lceil \log_2(d+1)\rceil}$. Thus setting $\alpha = \sqrt{Cd}\log\frac{\sqrt{C}}{r}$ gives $\mathrm{Pr}_{\mathcal{Q}}[||x|| > \alpha] < e\sqrt{C}r$. Also, by Theorem 11 of [BL13], for sufficiently small non-negative $\beta_2 \leq \frac{1}{56\lceil \log_2(d+1)\rceil}$, we have $\mathrm{Pr}_Q[|w_{h^*} \cdot x| \leq cr\alpha] \leq c'r\sqrt{Cd}\log\frac{\sqrt{C}}{r}$ for constant $c'$. The proof is concluded by a union bound and applying Definition 9. $\qquad\square$

We further consider the case where the distributions belong to the broad class of isotropic $s$-concave distributions. In particular, unlike $\beta$-log-concave distributions, the distributions from this class can potentially be fat-tailed.

**Definition 17** ($s$-concavity). *A density function $f : \mathbb{R}^d \to \mathbb{R}_{\geq 0}$ is $s$-concave for $s \in (-\infty, 1] \cup \{-\infty\}$ if for any $\lambda \in [0, 1]$ and any $x_1, x_2 \in \mathbb{R}^d$, we have $f(\lambda x_1 + (1 - \lambda)x_2) \geq (\lambda f(x_1)^s + (1 - \lambda)f(x_2)^s)^{1/s}$.*

Note that any $s$-concave function is also $s'$-concave if $s > s'$. Moreover, concave functions are 1-concave and log-concave functions are $s$-concave for any $s < 0$. Using results from [BZ17], we adapt the arguments in Theorem G.1 to show a bound on the disagreement coefficient when $\mathcal{P}$ is isotropic $\beta$-log-concave and $\mathcal{Q}$ is isotropic $s$-concave.

**Theorem G.2.** *Let the concept space $\mathcal{H}$ be the class of linear separators in $\mathbb{R}^d$. Let $\mathcal{P}$ be isotropic $\beta$-log-concave and $\mathcal{Q}$ be isotropic $s$-concave, over $\mathbb{R}^d$. Then for $s \geq -1/(2d + 3)$ and sufficiently small non-negative $\beta \leq \frac{1}{56\lceil \log_2(d+1)\rceil}$, we have $\Theta_{\mathcal{P}\to\mathcal{Q}}(\varepsilon) = O\left(\sqrt{d}\frac{2(1+ds)^2}{s+s^2(d+2)}(1 - \varepsilon^{s/(1+ds)})\right)$.*

*Proof.* Similar to the proof of Theorem G.1, we can apply the whitening transform from Theorem 16 of [BL13] provided $(1/20 + c_1)\sqrt{1/C_1 - c_1^2} \leq 1/9$, where $C_1 = e^{\beta \lceil \log_2(d+1)\rceil}$ and $c_1 = e(C_1 - 1)\sqrt{2C_1}$. It may be verified that this condition holds for $0 \leq \beta \leq \frac{1}{56\lceil \log_2(d+1)\rceil}$. We can also show that the set $\{x \mid ||x|| > \alpha\} \cup \{x \mid |w_{h^*} \cdot x| \leq cr\alpha\}$ contains the disagreement region $\mathrm{DIS}(B_{\mathcal{P}}(h^*, r)))$.

By Theorem 11 of [BZ17], we have $\mathrm{Pr}_{\mathcal{Q}}[|w_{h^*} \cdot x| \leq cr\alpha] \leq \frac{2(1+ds)}{1+s(d+2)} \cdot cr\alpha$. By Theorem 5 of [BZ17], since $\mathcal{Q}$ is an isotropic $s$-concave distribution, we have $\mathrm{Pr}_Q[||x|| > t\sqrt{d}] < \left(1 - \frac{c_1 st}{1+ds}\right)^{(1+ds)/s}$, for any $t \geq 16$ and absolute constant $c_1$. This implies $\mathrm{Pr}_{\mathcal{Q}}[||x|| > c_1\sqrt{d}\frac{1+ds}{s}(1 - r^{s/(1+ds)})] < Cr$ for some constant $C$. Thus setting $\alpha = c_1\sqrt{d}\frac{1+ds}{s}(1 - r^{s/(1+ds)})$ gives $\mathrm{Pr}_{\mathcal{Q}}[||x|| > \alpha] < Cr$. Also, for this $\alpha$, we have $\mathrm{Pr}_{\mathcal{Q}}[|w_{h^*} \cdot x| \leq cr\alpha] \leq c\frac{2(1+ds)}{1+s(d+2)} \cdot c_1\sqrt{d}\frac{1+ds}{s}(1 - r^{s/(1+ds)})r = c'\sqrt{d}\frac{2(1+ds)^2}{s+s^2(d+2)}(1 - r^{s/(1+ds)})r$ for constant $c'$. The proof is concluded by a union bound and applying Definition 9. $\qquad\square$

**Smooth classification boundaries.** We also illustrate our notion for more general concept spaces beyond linear separators. Specifically, we consider classifiers with smooth boundaries (Definition 15). If we further assume that the probability density may be upper and lower bounded by an $\alpha$th order smooth function, we can bound the disagreement coefficient for shift from $\mathcal{P}$ to $\mathcal{Q}$. Interestingly, while [Wan11] need the distribution to be sandwiched between smooth functions, our result only needs a lower bound on the smoothness of $\mathcal{P}$ and an upper bound on the smoothness of $\mathcal{Q}$.

**Theorem G.3.** *Let the instance space be $\mathcal{X} = [0,1]^{d+1}$. Let the hypothesis space be $\mathcal{H}_\alpha^C$, with $d < \alpha < \infty$. If the marginal distributions $\mathcal{P}_\mathcal{X}, \mathcal{Q}_\mathcal{X}$ have densities $p(x)$ and $q(x)$ on $[0,1]^{d+1}$ such that there exists an $\alpha$th order smooth function $g(x)$ and $a_p, b_q \in \mathbb{R}_+$ such that $a_p g(x) \leq p(x)$ and $q(x) \leq b_q g(x)$ for all $x \in [0,1]^{d+1}$, then $\Theta_{\mathcal{P} \to \mathcal{Q}}(\varepsilon) = O\left(b_q a_p^{-\frac{\alpha}{d+\alpha}} \varepsilon^{-\frac{d}{d+\alpha}}\right)$.*

*Proof.* We will extend the arguments from [Wan11] to the distribution shift setting. Let $\boldsymbol{x} = (x_1, \ldots, x_d) \in [0,1]^d$ and let $h \in B_P(h^*, r)$ where $h^* \in \mathcal{H}_\alpha$ is the target concept. Denote $\Phi_h^p(\boldsymbol{x}) = \int_{f_{h^*}(\boldsymbol{x})}^{f_h(\boldsymbol{x})} p(\boldsymbol{x}, x_{d+1}) dx_{d+1}$, and $\Phi_h^q(\boldsymbol{x}) = \int_{f_{h^*}(\boldsymbol{x})}^{f_h(\boldsymbol{x})} q(\boldsymbol{x}, x_{d+1}) dx_{d+1}$. It is easy to verify by taking derivatives that $\tilde{\Phi}_h(\boldsymbol{x}) = \int_{f_{h^*}(\boldsymbol{x})}^{f_h(\boldsymbol{x})} g(\boldsymbol{x}, x_{d+1}) dx_{d+1}$ is $\alpha$th order smooth. Since $h \in B_P(h^*, r)$,

$$\int_{[0,1]^d} |\tilde{\Phi}_h(\boldsymbol{x})| d\boldsymbol{x} \leq \int_{[0,1]^d} \frac{1}{a_p} |\Phi_h^p(\boldsymbol{x})| d\boldsymbol{x} \leq \frac{r}{a_p}.$$

By Lemma 11 of [Wan11], this implies $||\tilde{\Phi}_h||_\infty = O\left(\left(\frac{r}{a_p}\right)^{\frac{\alpha}{d+\alpha}}\right)$, and therefore $||\Phi_h^q||_\infty \leq b_q ||\tilde{\Phi}_h||_\infty = O\left(b_q a_p^{-\frac{\alpha}{d+\alpha}} r^{\frac{\alpha}{d+\alpha}}\right)$. Since this holds for any $h \in B_\mathcal{P}(h^*, r)$, we have

$$\sup_{h \in B_P(h^*, r)} ||\Phi_h^q||_\infty = O\left(b_q a_p^{-\frac{\alpha}{d+\alpha}} r^{\frac{\alpha}{d+\alpha}}\right).$$

By definition of region of disagreement, we have

$$\Pr_{x \sim \mathcal{Q}_\mathcal{X}}[x \in DIS(B_\mathcal{P}(h^*, r))] = \Pr_{x \sim \mathcal{Q}_X}[x \in \cup_{h \in B_\mathcal{P}(h^*, r)}\{x' \mid h(x') \neq h^*(x')\}]$$

$$\leq 2 \int_{[0,1]^d} \sup_{h \in B_\mathcal{P}(h^*, r)} ||\Phi_h^q||_\infty d\boldsymbol{x}$$

$$= O\left(b_q a_p^{-\frac{\alpha}{d+\alpha}} r^{\frac{\alpha}{d+\alpha}}\right).$$

The result follows from definition of $\Theta_{\mathcal{P} \to \mathcal{Q}}(\epsilon)$. $\qquad\square$

# H   Simple examples for the $\mathcal{P} \to \mathcal{Q}$ disagreement coefficient

**Example 1.** (Non-overlapping spheres the same center) Let $\mathcal{P}, \mathcal{Q}$ be uniform distribution over a sphere with the center at the origin with radius 1 and 2 respectively. Let $\mathcal{H}$ be a class of linear separators that pass through the origin and $h^* \in \mathcal{H}$. By symmetry, we have

$$\Theta_{\mathcal{P} \to \mathcal{Q}}(\varepsilon) = \sup_{r \geq \varepsilon} \frac{\Pr_\mathcal{Q}(\text{DIS}(B_\mathcal{P}(h^*, r)))}{r}$$

$$= \sup_{r \geq \varepsilon} \frac{\Pr_\mathcal{P}(\text{DIS}(B_\mathcal{P}(h^*, r)))}{r}$$

$$= \Theta_\mathcal{P}(\varepsilon).$$

The disagreement coefficient from $\mathcal{P}$ to $\mathcal{Q}$ is the same as the disagreement coefficient on $\mathcal{P}$.

**Example 2.** (Thresholds) Let $\mathcal{P}, \mathcal{Q}$ be uniform distribution over an interval $[-\frac{1}{2}, \frac{1}{2}]$ and $[-1, 1]$ respectively. Let $\mathcal{H}$ be a class of a threshold function. Let $h^*$ have a thereshold at 0. We have $\text{DIS}(B_\mathcal{P}(h^*, r)) = [-r, r]$ and

$$\Theta_{\mathcal{P} \to \mathcal{Q}}(\varepsilon) = \sup_{r \geq \varepsilon} \frac{\Pr_\mathcal{Q}([-r, r])}{r}$$

$$= \frac{\frac{2r}{2}}{r} = 1$$

compared to the disagreement coefficient $\Theta_\mathcal{P}(\varepsilon) = 2$.

# I  Additional proof details for distribution shift

*Proof.* (of Theorem 5.1) If $S \sim \mathcal{P}^m$, with $m \geq \frac{c}{\epsilon^2}(d + \ln\frac{1}{\delta})$ for some sufficiently large constant $c$, we have by uniform convergence ([AB99], Theorem 4.10) that with probability at least $1 - \delta$, we have $d_{\mathcal{P}}(h, h^*) \leq d_S(h, h^*) + \epsilon$ for all $h \in \mathcal{H}$. Here $d_{\mathcal{D}}(h_1, h_2) = \Pr_{x \sim \mathcal{D}_X}[h_1(x) \neq h_2(x)]$, and $d_S(h_1, h_2) = \frac{1}{|S|}\sum_{x \in S} \mathbb{I}[h_1(x) \neq h_2(x)]$. Therefore, $\mathrm{Agree}(B_{\mathcal{P}}^{\mathcal{H}}(h^*, \epsilon)) \subseteq \mathrm{Agree}(\mathcal{H}_0(S)) \subseteq R^{\mathcal{L}}(S)$ in this event. Denoting this event by '$E$' and its complement by '$\bar{E}$', we have

$$
\begin{aligned}
\Pr_{x \sim \mathcal{Q}, S \sim \mathcal{P}^m}[x \in R^{\mathcal{L}}(S)] &= \Pr_{x \sim \mathcal{Q}}[x \in R^{\mathcal{L}}(S) \mid E]\Pr[E] + \Pr_{x \sim \mathcal{Q}}[x \in R^{\mathcal{L}}(S) \mid \bar{E}]\Pr[\bar{E}] \\
&\geq \Pr_{x \sim \mathcal{Q}}[x \in R^{\mathcal{L}}(S) \mid E] \cdot (1 - \delta) \\
&\geq \Pr_{x \sim \mathcal{Q}}[x \in R^{\mathcal{L}}(S) \mid E] - \delta \\
&\geq \Pr_{x \sim \mathcal{Q}}[x \in \mathrm{Agree}(B_{\mathcal{P}}^{\mathcal{H}}(h^*, \epsilon))] - \delta.
\end{aligned}
$$

Noting $\Pr_{x \sim \mathcal{Q}}[x \in \mathrm{Agree}(B_{\mathcal{P}}^{\mathcal{H}}(h^*, \epsilon))] = 1 - \Pr_{x \sim \mathcal{Q}}[x \in \mathrm{DIS}(B_{\mathcal{P}}^{\mathcal{H}}(h^*, \epsilon))]$ and using Definition 9 completes the proof. $\square$

# J  Safely-reliable correctness under distribution shift

There is a growing practical [SSZ+20, SIE+20] as well as recent theoretical interest [DGH+23] in the setting of 'robustness transfer', where one simultaneously expects adversarial test-time attacks as well as distribution shift. We will study the reliability aspect for this more challenging setting. We note that the definition of a robustly-reliable learner does not depend on the data distribution (see Definition 3) as the guarantee is pointwise. Our optimality result in Section 3 applies even when a test point is drawn from a different distribution $\mathcal{Q}$. In this case, the safely-reliable region instead would have a different probability mass.

**Definition 18** ($\mathcal{P} \to \mathcal{Q}$ safely-reliable correctness). *The $\mathcal{P} \to \mathcal{Q}$ safely-reliable correctness of $\mathcal{L}$ (at sample rate $m$, for distribution shift from $\mathcal{P}$ to $\mathcal{Q}$, w.r.t. robust loss $\ell$) is defined as the probability mass of its safely-reliable region under Q, on a sample $S \sim \mathcal{P}^m$, i.e. $PQR_\ell^{\mathcal{L}}(S, \eta_1, \eta_2) := \Pr_{x \sim Q, S \sim P^m}[x \in SR_\ell^{\mathcal{L}}(S, \eta_1, \eta_2)]$.*

We will now combine our results on test-time attacks and distribution shift to give a general bound on the $\mathcal{P} \to \mathcal{Q}$ safely-reliable correctness for the different robust losses (Definition 1).

**Theorem J.1.** *Let $\mathcal{Q}$ be a realizable distribution shift of $\mathcal{P}$ with respect to $\mathcal{H}$, and $h^* \in \mathcal{H}$ be the target concept. There exist learners for robust losses $\ell_{CA}, \ell_{TL}, \ell_{ST}, \ell_{IA}$ with $\mathcal{P} \to \mathcal{Q}$ safely-reliable correctness given by*

*(a)* $PQR_{CA}^{\mathcal{L}}(S, \eta_1, \eta_2) = \Pr_{x \sim Q}[\mathbf{B}_{\mathcal{M}}(x, \eta_1) \cap \{z \mid h^*(z) = h^*(x)\} \subseteq \mathrm{Agree}(\mathcal{H}_0(S))]$,

*(b)* $PQR_{TL}^{\mathcal{L}}(S, \eta_1, \eta_2) = \Pr_{x \sim Q}[\mathbf{B}_{\mathcal{M}}(x, \eta_1) \subseteq \mathrm{Agree}(\mathcal{H}_0(S))]$,

*(c)* $PQR_{ST}^{\mathcal{L}}(S, \eta_1, \eta_2) = \Pr_{x \sim Q}[\mathbf{B}_{\mathcal{M}}(x, \eta_1) \subseteq \{z \mid \mathbf{B}^o(z, \eta) \subseteq \mathrm{Agree}(\mathcal{H}_0(S)) \wedge h(x) = h(z), \forall x \in \mathbf{B}^o(z, \eta), h \in \mathcal{H}_0(S)\}]$,

*(d)* $PQR_{IA}^{\mathcal{L}}(S, \eta_1, \eta_2) = \Pr_{x \sim Q}[(\{z \mid \mathbf{B}(z, \eta_2) \subseteq \mathrm{Agree}(\mathcal{H}_0(S))\} \cap \{z \mid h^*(z) = 1\}) \cup (\{z \mid \mathbf{B}(z, \eta_1) \subseteq \mathrm{Agree}(\mathcal{H}_0(S))\} \cap \{z \mid h^*(z) = 0\})]$.

*Proof.* The proof follows by applying Theorems B.1 and B.2, and using Definitions 6 and 18. $\square$

We consider an example when the training distribution $\mathcal{P}$ is isotropic log-concave and the test distribution $\mathcal{Q}_\mu$ is log-concave with its mean shifted by $\mu$ but the covariance matrix is still an identity matrix (see Figure 4, right).

**Theorem J.2.** *Let $\mathcal{P}, \mathcal{Q}$ be isotropic log-concave over $\mathbb{R}^d$. Let $\mathcal{Q}_\mu$ be a distribution after shifting the mean of $\mathcal{Q}$ by $\mu \in \mathbb{R}^d$. Let $\mathcal{H} = \{h : x \to \mathrm{sign}(\langle w_h, x \rangle) \mid w_h \in \mathbb{R}^d, \|w_h\|_2 = 1\}$ be the class of linear separators. Let $\mathbf{B}(\cdot, \eta)$ be a $L_2$ ball perturbation with radius $\eta$. For $S \sim \mathcal{P}^m$, for $m = \mathcal{O}(\frac{1}{\varepsilon^2}(\mathrm{VCdim}(\mathcal{H}) + \ln\frac{1}{\delta}))$, for an optimal robustly-reliable learner $\mathcal{L}$, we have*

(a) $\mathrm{Pr}_{\mathcal{Q}_\mu}(SR_{TL}^{\mathcal{L}}(S, \eta_1, \eta_2)) \geq 1 - 2(\eta_1 + \|\mu\|_2) - \tilde{\mathcal{O}}(\sqrt{d}\varepsilon)$ *with probability* $1 - \delta$,

(b) $SR_{CA}^{\mathcal{L}}(S, \eta_1, \eta_2) = SR_{TL}^{\mathcal{L}}(S, \eta_1, \eta_2)$,

(c) $\mathrm{Pr}_{\mathcal{Q}_\mu}(SR_{ST}^{\mathcal{L}}(S, \eta_1, \eta_2)) \geq 1 - 2(\eta_1 + \eta_2 + \|\mu\|_2) - \tilde{\mathcal{O}}(\sqrt{d}\varepsilon)$ *with probability* $1 - \delta$,

(d) $\mathrm{Pr}_{\mathcal{Q}_\mu}(SR_{IA}^{\mathcal{L}}(S, \eta_1, \eta_2)) \geq 1 - (\eta_1 + \eta_2 + 2\|\mu\|_2) - \tilde{\mathcal{O}}(\sqrt{d}\varepsilon)$ *with probability* $1 - \delta$.

*The $\tilde{\mathcal{O}}$-notation suppresses dependence on logarithmic factors and distribution-specific constants.*

*Proof.* From triangle inequality, we know that $\mathbf{B}(x - \mu, r + \|\mu\|_2) \supseteq \mathbf{B}(x, r)$. We can simply extend the proofs from Theorem 3.3. Recall that $SR_{CA}^{\mathcal{L}}(S, \eta_1, \eta_2) = SR_{TL}^{\mathcal{L}}(S, \eta_1, \eta_2) = \{x \mid \mathbf{B}(x, \eta_1) \subseteq \mathrm{Agree}(\mathcal{H}_0(S))\} \supseteq \{x \mid \mathbf{B}(x - \mu, \eta_1 + \|\mu\|_2) \subseteq \mathrm{Agree}(\mathcal{H}_0(S))\}$ and $SR_{ST}^{\mathcal{L}}(S, \eta_1, \eta_2) = \{x \in \mathcal{X} \mid \mathbf{B}_{\mathcal{M}}(x, \eta_1 + \eta_2) \subseteq \mathrm{Agree}(\mathcal{H}_0(S))\} \supseteq \{x \in \mathcal{X} \mid \mathbf{B}_{\mathcal{M}}(x - \mu, \eta_1 + \eta_2 + \|\mu\|_2) \subseteq \mathrm{Agree}(\mathcal{H}_0(S))\}$. When $x$ is drawn from a distribution $\mathcal{Q}_\mu$, we know that $x - \mu$ follows a distribution $\mathcal{Q}$ which is isotropic log-concave. We can apply Lemma D.1 to bound the probability mass of the safely-reliable region under $\mathcal{Q}_\mu$. Similarly, we can do the same for $SR_{IA}^{\mathcal{L}}(S, \eta_1, \eta_2)$. □

Similar bounds on reliability under robustness transfer may be given for linear separators under more general source or target distributions, including isotropic $\beta$-log-concave or $s$-concave distributions, as well as for concept classes with smooth classification boundaries, by applying Theorem J.1 to the examples from previous sections.

# K  Agnostic setting

*Proof of Theorem 7.1.* The robustly-reliable learner $\mathcal{L}$ is given as follows. Set $h_S^{\mathcal{L}} = \mathrm{argmin}_{h \in \mathcal{H}} \mathrm{err}_S(h)$ i.e. an ERM over $S$, and $r_S^{\mathcal{L}}(z) = \infty$ if $z \in \mathrm{Agree}(\mathcal{H}_\nu(S))$, else $r_S^{\mathcal{L}}(z) = -1$. To study the robustly-reliable region, we assume there is some concept $h^* \in \mathcal{H}$ which satisfies $\mathrm{err}_S(h^*) \leq \nu$. By definition of ERM, $\mathrm{err}_S(h_S^{\mathcal{L}}) \leq \mathrm{err}_S(h^*) = \nu$, or $h_S^{\mathcal{L}} \in \mathcal{H}_\nu(S)$. We first show that $\mathcal{L}$ is robustly-reliable. For $z \in \mathcal{X}$, if $r_S^{\mathcal{L}}(z) = \eta > 0$, then $z \in \mathrm{Agree}(\mathcal{H}_\nu(S))$. We have $h^*(z) = h_S^{\mathcal{L}}(z)$ since the classifiers $h^*, h_S^{\mathcal{L}} \in \mathcal{H}_\nu(S)$ and $z$ lies in the agreement region of classifiers in $\mathcal{H}_\nu(S)$ in this case. Thus, we have $\ell_{CA}^{h^*}(h_S^{\mathcal{L}}, x, z) = 0$ for any $x$ such that $z \in \mathbf{B}_{\mathcal{M}}^o(x, \eta)$. In the $\eta = 0$ case, $h^*(z) = h_S^{\mathcal{L}}(z)$ by definiton and the same argument applies. Therefore, $\mathrm{RR}_{CA}^{\mathcal{L}}(S, \nu, \eta) \supseteq \mathrm{Agree}(\mathcal{H}_\nu(S))$ for all $\eta \geq 0$ follows from the setting $r_S^{\mathcal{L}}(z) = \infty$ if $z \in \mathrm{Agree}(\mathcal{H}_\nu(S))$.

Conversely, let $z \in \mathrm{DIS}(\mathcal{H}_\nu(S))$. There exist $h_1, h_2 \in \mathcal{H}_\nu(S)$ such that $h_1(z) \neq h_2(z)$. By definition, robustly-reliable learning with $\eta = 0$ is not possible for $z$. If possible, let there be a robustly-reliable learner $\mathcal{L}$ such that $z \in \mathrm{RR}_{CA}^{\mathcal{L}}(S, \nu, \eta)$ for some $\eta > 0$. By definition of the robust-reliability region, we must have $r_S^{\mathcal{L}}(z) > 0$. By definition of a ball, we have $z \in \mathbf{B}_{\mathcal{M}}^o(z, \eta)$ for any $\eta > 0$, and therefore $\ell_{CA}^{h^*}(h_S^{\mathcal{L}}, z, z) = 0$ for every $h^* \in \mathcal{H}$ such that $\mathrm{err}_S(h^*) \leq \nu$. But then we must have $h_S^{\mathcal{L}}(z) = h^*(z)$ by definition of $\ell_{CA}$. But we can set $h^* = h_1$ or $h^* = h_2$ since both are in $\mathcal{H}_\nu(S)$. But $h_1(z) \neq h_2(z)$, and therefore $h_S^{\mathcal{L}}(z) \neq h^*(z)$ for one of the above choices for $h^*$, contradicting that $\mathcal{L}$ is robustly-reliable. □