# OpenReview forum: "Reliable learning in challenging environments"
_NeurIPS.cc/2023/Conference — NeurIPS 2023 poster_

### Official Review · Reviewer_nCeL · 2023-06-30

**Soundness:** 3 good
**Presentation:** 3 good
**Contribution:** 3 good
**Rating:** 5
**Confidence:** 3

**Summary:**

The paper develops learning methods that provide theoretical guarantees for point-wise predictions in challenging scenarios at test-time. In particular, the methods presented address situations affected by adversarial attacks and distribution shifts. The paper provides a theoretical contribution to a relevant line of work on reliable learning.

**Strengths:**

The paper presents a significant theoretical contribution on reliable learning, addressing important scenarios such as those affected by adversarial attacks and distribution shifts. In addition, the paper covers several different notions of losses and reliability. The results presented can lead to a better understanding of the possibilities and limitations of such scenarios.

**Weaknesses:**

The main limitation I can see in the paper is the reliance on the realizable case. I guess that reliance cannot be easily avoided but it may be worth to further discuss and emphasize that issue. For instance, the guarantees in the paper are contingent to the fact that we are in a realizable setting, which is not possible to know in most practical cases. Such assumption is even more significant for the distribution shift case, since it constraints the possible shifts.

The paper contains many theoretical results for multiple losses, scenarios, etc. In general, it is good to have many interesting results but the paper is quite dense, and it is difficult to get the main ideas. For instance, in Section 4, the authors mention "The optimal robustly-reliable learner described above may be implemented" but the authors hardly describe such learner before Section 4 besides stating that it exists (it is described in the proofs). The paper would be significantly more readable (at least for readers not very familiar with the topic) if the authors first describe the main ideas and results over the simplest scenario and then generalize them to other cases.

It is not clear that the probabilities lower bounded in Theorem 3.3. are not very small or that the abstention probabilities are not very high. I think this point is important because the learners would not be very useful in other cases. It would be good if the authors can quantify numerically those probabilities. The bounds provided show that the probabilities increase when the number of samples increases and the dimension decreases but it is not clear in what scenarios those probabilities are sizable or even the lower bound is larger than 0.


**Questions:**


Why is that the robustly-reliable region does not contain the instances x such that r(x)=0?

I guess there is a typo in Theorem 3.1 and the subscript CA in "RR^\set{L}_CA(S,\eta)" should be omitted as in Definition 4 or be a generic W?

English usage can be improved in few places, e.g., "Given a 0-1 loss function" should read "Given the 0-1 loss function", "provide bounds the probability mass" should read "provide bounds for the probability mass"


**Limitations:**

The authors correctly describe the limitations of the results in the paper. In that sense, it would be good if the reliance on the realizable case is further discussed.

---

> ### Author Rebuttal · Authors · 2023-08-10
>
> **Weaknesses**
>
> > 1. The main limitation I can see in the paper is the reliance on the realizable case. I guess that reliance cannot be easily avoided but it may be worth to further discuss and emphasize that issue. For instance, the guarantees in the paper are contingent to the fact that we are in a realizable setting, which is not possible to know in most practical cases. Such assumption is even more significant for the distribution shift case, since it constraints the possible shifts.
>
> We believe that it is always good to first address the realizable case to build the intuition for the definitions and results.  This follows a well-established pattern of developing theories in the learning theory literature.  As we discuss below, extension to the non-realizable (agnostic) setting is possible, building from the definitions and concepts we have developed in the realizable case.
> In fact, it is possible to relax the realizability assumption for the robustly-reliable learner. We will provide a discussion in the camera-ready version. We include a definition of an agnostic robustly-reliable learner below, along with the statement of a theorem for bounding the robustly-reliable region for the true label robust loss function. Our result below implies that the robustly-reliable region in the agnostic case is a slightly smaller agreement region than in the realizable case, depending on the error rate $\nu$ of the best classifier in $\mathcal{H}$. Below, we will illustrate the definition and result in the agnostic case for the TL loss.
>
> **Definition**: A learner $\mathcal{L}$ is *$\nu$-tolerably* robustly-reliable w.r.t. $\mathcal{M}$-ball attacks for sample $S$, hypothesis space $\mathcal{H}$ and robust loss function $\ell$ if, for every concept $h^*\in\mathcal{H}$ with $\mathrm{err}\_{S}(h^*)\le\nu$, the learner outputs functions $h^\mathcal{L}\_{S}:\mathcal{X} \to \mathcal{Y}$ and $r^\mathcal{L}\_{S}:\mathcal{X} \to [0,\infty)\cup\\{-1\\}$ such that for all $x,z\in \mathcal{X}$ if $r^\mathcal{L}\_{S}(z)=\eta > 0$ and $z\in \mathbf{B}^o\_\mathcal{M}(x,\eta)$ then $\ell^{h^*}(h^\mathcal{L}\_{S},x,z)=0$. Further, if $r^\mathcal{L}_{S}(z) = 0$, then $h^*(z)=h^\mathcal{L}\_{S}(z)$.
>
> Given sample $S$ such that some concept $h^*\in\mathcal{H}$ satisfies $\mathrm{err}\_S(h^*)\le \nu$, the robustly-reliable region of $\mathcal{L}$ is defined as $RR^{\mathcal{L}}(S,\nu,\eta)=\\{x\in\mathcal{X}\mid r^\mathcal{L}\_{S}(x) \ge \eta\\}$ for $\nu,\eta\ge 0$.
>
> We can prove the following theorem (stated below for the true label loss $\ell\_{\text{TL}}$) which gives pointwise optimal bounds on the robustly-reliable region.
>
> **Theorem**: Let $\mathcal{H}$ be any hypothesis class with respect to $\mathcal{M}$-ball attacks and robust loss function $\ell\_{\text{TL}}$, for $\eta\ge 0$,
>  - There exists a robustly-reliable learner $\mathcal{L}$ such that $RR^{\mathcal{L}}\_{\text{TL}}(S,\nu,\eta)\supseteq \text{Agree}(\mathcal{H}\_\nu(S))$,
> - For any robustly-reliable learner $\mathcal{L}$, $RR^{\mathcal{L}}\_{\text{TL}}(S,\nu,\eta)\subseteq \text{Agree}(\mathcal{H}\_\nu(S))$
>
> where $\mathcal{H}\_\nu(S) = \\{h \in \mathcal{H} \mid \mathrm{err}\_S(h)\le \nu\\}$. Furthermore, the safely-reliable region for robust loss function $TL$ is defined as $SR\_{TL}^{\mathcal{L}}(S,\nu,\eta_1,\eta_2)=\\{x\in\mathcal{X}\mid \mathbf{B}\_\mathcal{M}(x,\eta_1)\subseteq RR\_{TL}^{\mathcal{L}}(S,\nu,\eta_2)\\}$. Thus, the above theorem implies bounds on the safely-reliable region as well.
> We will include the above result, along with formal proof and discussion in the camera-ready version.
>
> > 2. The paper contains many theoretical results for multiple losses, scenarios, etc. In general, it is good to have many interesting results but the paper is quite dense, and it is difficult to get the main ideas. For instance, in Section 4, the authors mention "The optimal robustly-reliable learner described above may be implemented" but the authors hardly describe such learner before Section 4 besides stating that it exists (it is described in the proofs).
>
> Thank you for your point. Indeed, the optimal learner is provided in the proof of Theorem 3.1 in Appendix. We will describe the learner in Section 3 in the camera-ready version.
>
>
> > 3. It is not clear that the probabilities lower bounded in Theorem 3.3. are not very small or that the abstention probabilities are not very high. I think this point is important because the learners would not be very useful in other cases. It would be good if the authors can quantify numerically those probabilities. The bounds provided show that the probabilities increase when the number of samples increases and the dimension decreases but it is not clear in what scenarios those probabilities are sizable or even the lower bound is larger than 0.
>
> The probability $1 - \delta$ in the lower bound in Theorem 3.3 can be arbitrarily close to 1 as the number of sample m is large. Also, the non-abstention probability will be arbitrarily close to 1 for large sample size m. For the safely-reliable region, this non-abstention rate depends on the robust loss, but is generally large for small $\eta_1,\eta_2$. For example, it can be close to $1 - 2(\eta_1 + \eta_2)$ for the stability loss.
>
> Since our main purpose for the paper is developing theory, we would defer the numerical quantification of those probabilities in this simple setting and the real-world setting to future application-oriented research.
>
> **Questions**
> 1. The robustly-reliable region also includes $r(x) = 0$ which is when we know that the prediction is correct. In the definition 3, we have two cases when $r(x) = \eta > 0$  and $r(x) = 0$ for the mathematical rigorousness of the definition.
> 2. Yes, this is a typo. Thank you for pointing it out.
> 3. Thank you for your point, we will correct the grammar accordingly.

---

> > ### Comment · Reviewer_nCeL · 2023-08-15
> >
> > I thank the authors for their answers. I believe the paper deserves to be published and it will be improved in the camera ready version. Just an additional comment. I agree with the authors that starting with the realizable case is in general good to get the main intuition. However, for this paper I think it would be useful if the authors discuss such assumption in relation with the distribution shifts, since the realizable assumption somehow constraints the shifts addressed in the paper.

---

### Official Review · Reviewer_8X7Z · 2023-06-30

**Soundness:** 4 excellent
**Presentation:** 4 excellent
**Contribution:** 3 good
**Rating:** 7
**Confidence:** 3

**Summary:**

The authors explore advesarial test-time attacks and distribution shift. They propose a learning algorithm with performance guarantees.


**Strengths:**

The paper is well written with many intuitive illustrations.
The authors use refusal as a means to guarantee reliability. As noted by the authors, the trivial classifier that always refuses is reliable. Therefore, I particularly liked section 3.1, which characterizes the probability of refusal (or non-refusal). The concept of reliability radius is readily understandable.

The algorithms provided in section 4 can be efficient enough for many real-world applications.


**Weaknesses:**

The authors' analysis is limited to the realizable case; that the true target function is a member of the hypothesis class. While this simplifies analysis, it limits the applicability of the authors' results since in most real-world applications we do not know whether this assumption is valid.



**Questions:**

Can slack be added to the quadratic program in section 4 in the same manner as it is done for SVMs?

For the regularized objective in section 4, what good is a lower bound on the reliability radius? It can guarantee an unreliable prediction is unreliable, but cannot guarantee a reliable prediction is reliable. Yet it's the later case that is needed to not refuse for reliable test points.


**Limitations:**

Limitations are adequately addressed

---

> ### Author Rebuttal · Authors · 2023-08-10
>
> **Weaknesses**
>
> > The authors' analysis is limited to the realizable case; that the true target function is a member of the hypothesis class. While this simplifies analysis, it limits the applicability of the authors' results since in most real-world applications we do not know whether this assumption is valid.
>
> Thank you for your feedback. We agree that the realizable assumption may not hold in most real-world applications.  Starting by studying the realizable case helps to build intuition for what definitions and results are appropriate and possible in general.  This follows a well-established pattern of developing theories in the learning theory literature.  Fortunately, as we discuss below, extension to the non-realizable (agnostic) setting is possible, building from the definitions and concepts we have developed in the realizable case.
> We illustrate this for the true label loss $\ell_{\text{TL}}$ loss below, and remark this can be similarly done for the other loss functions studied in our work.
>
> **Definition**: A learner $\mathcal{L}$ is *$\nu$-tolerably* robustly-reliable w.r.t. $\mathcal{M}$-ball attacks for sample $S$, hypothesis space $\mathcal{H}$ and robust loss function $\ell$ if, for every concept $h^*\in\mathcal{H}$ with $\mathrm{err}\_{S}(h^*)\le\nu$, the learner outputs functions $h^\mathcal{L}\_{S}:\mathcal{X} \to \mathcal{Y}$ and $r^\mathcal{L}\_{S}:\mathcal{X} \to [0,\infty)\cup\\{-1\\}$ such that for all $x,z\in \mathcal{X}$ if $r^\mathcal{L}\_{S}(z)=\eta > 0$ and $z\in \mathbf{B}^o\_\mathcal{M}(x,\eta)$ then $\ell^{h^*}(h^\mathcal{L}\_{S},x,z)=0$. Further, if $r^\mathcal{L}_{S}(z) = 0$, then $h^*(z)=h^\mathcal{L}\_{S}(z)$.
>
> Given sample $S$ such that some concept $h^*\in\mathcal{H}$ satisfies $\mathrm{err}\_S(h^*)\le \nu$, the robustly-reliable region of $\mathcal{L}$ is defined as $RR^{\mathcal{L}}(S,\nu,\eta)=\\{x\in\mathcal{X}\mid r^\mathcal{L}\_{S}(x) \ge \eta\\}$ for $\nu,\eta\ge 0$.
>
> We can prove the following theorem (stated below for the true label loss $\ell\_{\text{TL}}$) which gives pointwise optimal bounds on the robustly-reliable region.
>
> **Theorem**: Let $\mathcal{H}$ be any hypothesis class with respect to $\mathcal{M}$-ball attacks and robust loss function $\ell\_{\text{TL}}$, for $\eta\ge 0$,
>  - There exists a robustly-reliable learner $\mathcal{L}$ such that $RR^{\mathcal{L}}\_{\text{TL}}(S,\nu,\eta)\supseteq \text{Agree}(\mathcal{H}\_\nu(S))$,
> - For any robustly-reliable learner $\mathcal{L}$, $RR^{\mathcal{L}}\_{\text{TL}}(S,\nu,\eta)\subseteq \text{Agree}(\mathcal{H}\_\nu(S))$
>
> where $\mathcal{H}\_\nu(S) = \\{h \in \mathcal{H} \mid \mathrm{err}\_S(h)\le \nu\\}$. Furthermore, the safely-reliable region for robust loss function $TL$ is defined as $SR\_{TL}^{\mathcal{L}}(S,\nu,\eta_1,\eta_2)=\\{x\in\mathcal{X}\mid \mathbf{B}\_\mathcal{M}(x,\eta_1)\subseteq RR\_{TL}^{\mathcal{L}}(S,\nu,\eta_2)\\}$. Thus, the above theorem implies bounds on the safely-reliable region as well.
>
>
> We will include the above result, along with formal proof and discussion in the camera-ready version.
>
> **Questions:**
> > Can slack be added to the quadratic program in section 4 in the same manner as it is done for SVMs?
>
> Yes, but with the slack, the optimal solution might not lie in the agreement region.
>
> > For the regularized objective in section 4, what good is a lower bound on the reliability radius? It can guarantee an unreliable prediction is unreliable, but cannot guarantee a reliable prediction is reliable. Yet it's the later case that is needed to not refuse for reliable test points.
>
> We can use the lower bound for the reliability guarantee. However, providing the bound on the difference between the lower bound and the actual reliability radius would be an interesting future research direction.

---

> > ### Comment · Reviewer_8X7Z · 2023-08-14
> > **Acknowledgement**
> >
> > I have read the authors' rebuttal. I feel they have adequately addressed my questions and concerns. I am leaning toward leaving my score as it is.

---

### Official Review · Reviewer_uQQ5 · 2023-07-06

**Soundness:** 4 excellent
**Presentation:** 3 good
**Contribution:** 3 good
**Rating:** 7
**Confidence:** 3

**Summary:**

This paper presents robustly-reliable learners with optimal guarantees for environments where the training and test data are not drawn from the same distribution, e.g., natural distribution shift and adversarial attacks during test time. The main idea is that for a given point, the robustly-reliable learner either outputs a prediction and a reliability region, or abstains from prediction. The prediction is guaranteed to be correct as long as the test-time perturbation is constrained to this reliability region.

**Strengths:**

* Designing reliable learners with guarantees for challenging environments is highly important and relevant to both the theoretical and applied machine learning community.
* It seems likely that other researchers will find relevant the reliability criterions and tools developed in the paper.
* The approach appears to be novel and technically sound. The claims of the paper are well-supported by extensive proofs.
* The submission is well organized and clearly written overall.


**Weaknesses:**


* A computationally efficient method is presented only for the case of linear separators. It is not clear how easily the presented tools can be used to obtain practical algorithms for more general cases (e.g., neural networks) in practice.
* An evaluation on a simple synthetic scenario to demonstrate the empirical effectiveness of the approach would make the paper more compelling. This is a minor point given that this is clearly a theory paper.


**Questions:**

Is there an efficient method for other types of classifiers and loss functions, such as neural networks?

**Limitations:**

The limitations are not discussed.

---

> ### Author Rebuttal · Authors · 2023-08-10
>
> > A computationally efficient method is presented only for the case of linear separators. It is not clear how easily the presented tools can be used to obtain practical algorithms for more general cases (e.g., neural networks) in practice.
> An evaluation on a simple synthetic scenario to demonstrate the empirical effectiveness of the approach would make the paper more compelling. This is a minor point given that this is clearly a theory paper.
>
> Thank you for your comment. We agree that a computationally efficient method for a general hypothesis class such as neural networks is an interesting direction. Since our main purpose for the paper is developing theory, we would defer the empirical evaluation of the methods and extension to neural networks to future application-oriented research.

---

> > ### Comment · Reviewer_uQQ5 · 2023-08-13
> >
> > Thank you for your reply. I read the other reviews and your responses to them. I lean towards keeping my score unchanged.

---

### Official Review · Reviewer_LDC7 · 2023-07-11

**Soundness:** 3 good
**Presentation:** 2 fair
**Contribution:** 2 fair
**Rating:** 4
**Confidence:** 3

**Summary:**

This paper studies the problem of classification under found different kinds of adversarial loss functions:
1) ST which I think is by far the most popular adversarial loss. This is same as the expected sup loss of [Madry et al. 2018](https://arxiv.org/abs/1706.06083).
2) TL which is equivalent to the the "exact in the ball" risk of [Gordeau et al](https://www.jmlr.org/papers/volume22/20-285/20-285.pdf) or the "error region risk" of [Diochnos et al](https://proceedings.neurips.cc/paper_files/paper/2018/file/3483e5ec0489e5c394b028ec4e81f3e1-Paper.pdf)
3) IA which is the same as ST but where the adversary only perturbs points belonging to one of the two labels
4) CA loss which is the same as TL with the additional constraint that the true label does not change at the perturbed point.

For each type of loss, the paper establishes the "optimal robustly reliable region". Here, the classification is deemed reliable if the the classification is correct on the _perturbed ppoint_. The papers shows that these regions can be computed efficiently for simple hypothesis classes like linear classifiers. The paper also proves lower bounds on "safely-reliable region" for linear separators under log-concave distributions, where a point is classified safely reliably at $\eta_1, \eta_2$ levels if it can be reliably classified at $\eta_2$ level, even when perturbed in a ball of radius $\eta_1$.

Finally, the paper also lower bounds the amount by which the reliability region can change under a distribution shift from P to Q in terms of a $P\to Q$ disagreement coefficient.



**Strengths:**

- Generality: The results of the paper hold for 4 kinds of adversarial loss functions, each with different use-cases.
- Nice result on reliable learning under distribution shift: Theorem 5.3 proves a connection between reliable learning under distribution shift, and a previously studied notion called the disagreement coefficient. This connection appears interesting and non-trivial.

**Weaknesses:**

- Writing: The paper is very difficult to read. There are many new definitions but very few illustrations / examples. The paper seems to be written in a hurry. Some of the results that are listed under main contributions in section 1.1 only appear in the appendix. Section 6 is super short - it only presents one new definition, followed by an example satisfying the definition the details of which are relegated to the appendix.
- Limited contributions: The paper makes limited contributions on the "safely-reliable" notion of classification (Definition 6), which I think is much more important than the notion of "reliable" classification (Definition 3, 4). Theorem 3.3 on probability mass on reliable region only holds for the special case of linear separators with isotropic log-concave distributions. Section 6 on safely-reliable correctness under distribution shift establishes the safely-reliable correctness again only for the special case of linear separators with isotropic log-concave distributions. Further, the shifted distribution is also assumed to only shift in mean, while the covariance matrix remains as Identity matrix. Section 4 also focuses on simple settings like linear separators.

- Limited applicability of the notion of "reliability":  I want to stress again that the "safely-reliable" notion is more important than the "reliable" notion. The importance of safely-reliable region over the reliable-region is stated by the authors themselves in lines 216-218: "...we note that the probability mass of the robustly-reliable region may not be a meaningful way to quantify the overall reliability of a learner because a perturbation may lie outside of the support of the natural data distribution and have zero probability mass." Basically, a classifier can be robustly reliable at z with level $\eta$ even if predicts a different label at a slightly perturbed point x such that d(x, z) < $\eta$. Hence, reliability does not ensure robustness against adversarial attacks. Literature on certified robustness (for example, randomized smoothing, interval bound propagation) all focus on something like the "safely-reliable" notion rather than reliable. In lines 91-94, the paper says "Prior works have examined pointwise consistency guarantees [SKL17, CRK19, WLF22], i.e. the classifier’s prediction is guaranteed not to have changed under an attack. In contrast, we study a much more desirable property of reliability—guaranteeing that the prediction of the algorithm is correct." I would like to see the paper make a stronger case advocating for the study of reliability than the one given here.

**Questions:**

- Definition 7 is unclear: What is z in line 296? I think it should be any x' in U(x).
- I would recommend cutting down on the size of section 1.1: summary of contributions.
- It would be good to have more examples to accompany the definitions, for instance contrasting the reliable vs safely-reliable notions.

I would also like the authors to comment on my view of the paper in the "weaknesses" section.

**Limitations:**

Yes.

---

> ### Author Rebuttal · Authors · 2023-08-10
>
> **Weakness**
> > 1. Writing:
>
> We will use the extra page in the camera-ready version to bring to the main body some of the results that now appear in the Appendix. Concerning illustrations, we already have 4 illustrations in the main body and in fact, the other reviewers appreciated the clarity and the many examples. “The submission is well organized and clearly written overall.” (Reviewer uQQ5) and “The paper is well written with many intuitive illustrations.” (Reviewer 8X7Z).
>
> > 2. Limited contributions:
>
> We respectfully disagree on the limited contribution of our paper. Our main contributions are numerous and we are happy to clarify this in the camera-ready version. First of all, coming up with the right definitions to express our concepts, in particular,  robustly-reliable learner (Definition 3), robustly-reliable region (Definition 4) and safely-reliable region (Definition 6). Given these definitions, we work out the abstract results;  Theorem 3.1 provides an optimal characterization of the robustly-reliable region for any hypothesis class (which further implies optimality of the safely-reliable region); Theorem 5.1 for safely-reliable correctness is also applicable to general settings. Additionally, we have numerous examples. The result presented in the main paper such as in Theorem 3.3 is one concrete example that the safely-reliable region can be large.  We also have additional results for a much more general class of classifiers with smooth boundaries in Appendix E and G  for both safely-reliable region (implies bound on safely-reliable correctness) and reliability under distribution shift.
>
> In particular, we address individual points below.
>
> -  “The paper makes limited contributions on the "safely-reliable" notion of classification (Definition 6), which I think is much more important than the notion of "reliable" classification (Definition 3, 4).”
>
> We disagree with this characterization. Definition 6 implies that our tight characterization of the optimal robustly-reliable region extends to the safely-reliable region as they are closely related for each robust loss function. Moreover, our examples that compute the probability masses focus on the safely-reliable region instead of the robustly-reliable region (Theorem 3.3, Appendix E ).
>
> - “Theorem 3.3 on probability mass on reliable region only holds for the special case of linear separators with isotropic log-concave distributions.”
>
> We also provide bounds on the probability mass of the safely-reliable region for general smooth classifiers (Lines 263-264, Appendix E, Theorem E.3).
>
> -  “Section 6 on safely-reliable correctness under distribution shift establishes the safely-reliable correctness again only for the special case of linear separators with isotropic log-concave distributions.”
>
> By Definition 10, our previous results on the safely-reliable region of smooth boundary classifiers also extend to the “safely-reliable correctness” in Section 6. We mention the mean-shift example to highlight the advantage over previously studied measures like $\mathcal{H}$-divergence and the discrepancy distance.
>
> -  “Section 4 also focuses on simple settings like linear separators.”
>
> We extend beyond the linear separator to a wide range of hypothesis classes in Appendix F.
>
> > 3. Limited applicability of the notion of "reliability":
>
>
>
> We would like to clarify this major misunderstanding between the robustness and reliability guarantee. The prior work on certified robustness [SKL17, CRK19, WLF22] is different from our proposed notion of safely-reliable. The certified robustness guarantee is only that a prediction does not change with an adversarial perturbation, but it does not guarantee that the prediction is correct (neither for the original point nor the perturbation); in particular, a constant function is always certified robust but it may not be useful. In contrast, a robustly-reliable learner guarantees that, for any test point $x$ and perturbation $z$, if $z$ has distance less than $\eta$ to $x$ ($\eta$ = reliability radius), then the prediction will be “correct” (robust loss zero) in a sense informed by which robust loss we are addressing; we discuss this idea for several different losses, leading to different interpretations of this guarantee.  For the stability loss, the prediction being “correct” means that it predicts the true label of the original point $x$; in particular, this implies certified robustness, but is even stronger, since it also guarantees the correct label.  For the “true label” loss, being “correct” means that it predicts the true label of the perturbation $z$.  For the “constrained adversary” loss, being “correct” means predicting the true label for both $x$ and $z$ assuming the adversary would only perturb $x$ to $z$ if they have the same true label.
>
> To summarize, for a given robust loss function, a robustly-reliable region (Definition 4) is a region where points in this region are guaranteed to have a zero robust loss (to be “correct”). A safely-reliable region further guarantees that even after a perturbation of some distance $\eta_1$, the point is still in the robustly reliable region for some radius $\eta_2$.  So being safely-reliable means the algorithm is robust against adversaries trying to perturb a point $x$ to a $z$ where the learner is less “confident” (i.e., outputs a small reliability radius).
>
> **Questions**
>
> 1. Thank you for pointing this out, this is a typo. It is supposed to be $h(x) = h^*(x)$.
> 2. We will address this in the camera-ready version.
> 3. Thank you for your comment. The safely-reliable region is indeed the same as the reliable region when there is no adversarial perturbation. We will make this clearer in the final version.

---

> ### Comment · Senior_Area_Chairs · 2023-08-20
> **please take a look at the author response**
>
> Thank you.

---

### Decision · Program_Chairs · 2023-09-21

**Decision:**

Accept (poster)

**Comment:**

This interesting paper provides many new ideas in reliable learning; reviewers are uniformly positive and I'm glad to suggest acceptance, though I urge the authors to spend their additional camera ready page incorporating the detailed reviewer comments below.